# A scoping review of the use of traditional medicine for the management of ailments in West Africa

Selassi A. D'Almeida[1], Sahr E. Gbomor[2], Brima Osaio-Kamara[3], Mobolaji Timothy Olagunju[4], Olunike Rebecca Abodunrin[4], Morẹ́nikẹ́ Oluwátóyìn Foláyan[5,6]*

1 Universal Health Coverage Life Course Cluster, World Health Organisation, Freetown, Sierra Leone, 2 Department of Complementary and Alternative Medicine, Pharmacy Board of Sierra Leone, Freetown, Sierra Leone, 3 Directorate of Primary Health Care, Ministry of Health, Freetown, Sierra Leone, 4 Department of Epidemiology and Biostatistics, Nanjing Medical University, Jiangsu Province, China, 5 Department of Child Dental Health, Obafemi Awolowo University, Ile-Ife, Nigeria, 6 Oral Health Initiative, Nigerian Institute of Medical Research, Yaba, Lagos, Nigeria

* toyinukpong@yahoo.co.uk

## Abstract

### Background

The coexistence of traditional healing practices deeply rooted in cultural and historical contexts and the evolving landscape of modern healthcare approaches in West African societies creates a dynamic interplay between tradition and modernity in healthcare. This study aims to comprehensively map the landscape of traditional medicine use for health in West Africa.

### Methods

A scoping review was conducted following the Joanna Briggs Institute (JBI) methodology and reported according to the Preferred Reporting Items for Systematic Reviews and Meta-Analyses Extension for Scoping Reviews (PRISMA-ScR) guidelines. Research questions focused on the links between traditional medicine practices and health in West Africa. The systematic literature search covered PubMed, Web of Science, and CINAHL from database inception to September 2023. A descriptive analysis was conducted highlighting the years of publication, countries of publication, study designs of plant families and plant parts used for making traditional medicines, and the diseases the traditional remedies are for.

### Results

The search identified 3484 records, with 46 articles meeting the inclusion criteria. Publications spanned from 1979 to 2023, with no observed trend in the number of publications over successive decades. Nigeria had the highest number of publications (54.3%), followed by Ghana (19.6%). The studies employed various designs, including clinical trials, ethnobotanical, ethnopharmacological, and experimental designs. Plant families frequently studied included Combretaceae, Euphorbiaceae, and Rubiaceae. Traditional remedies address various health issues, highlighting their versatility, from general symptoms to specific diseases.

**Data Availability Statement:** All relevant data are within the manuscript and its Supporting Information files.

**Funding:** The author(s) received no specific funding for this work.

**Competing interests:** The authors have declared that no competing interests exist.

## Conclusion

This scoping review offers an extensive overview of traditional healing practices in West Africa. The studies highlighted in this review stress the necessity for culturally sensitive healthcare interventions. The widespread use of traditional medicine and the variety of practices underscore the importance of encouraging collaboration between traditional healers and modern healthcare professionals. This review also identifies knowledge gaps and areas needing further research, setting the stage for future exploration into West Africa's intricate healthcare landscape.

## Introduction

The dynamic interplay between tradition and modernity in healthcare practices remains a complex and intriguing facet of West African societies [1]. This interplay reflects the ongoing negotiation between longstanding traditional healing methods deeply rooted in cultural and historical contexts and the evolving landscape of modern healthcare approaches [2]. In West Africa, traditional healing practices often involve indigenous knowledge, rituals, and the use of natural remedies passed down through generations [3]. These practices coexist with, and at times, intersect with, modern medical interventions, technologies, and pharmaceuticals, thereby reflecting the adaptability and resilience of West African societies in navigating the complexities of health and well-being [4].

Patients often engage with traditional and modern healthcare systems, sometimes concurrently or sequentially, depending on their health needs, personal beliefs, and accessibility [5]. This health-seeking behaviors is a common practice across Africa, creating a nuanced healthcare ecosystem in the region [6]. This underscores the importance of acknowledging diverse healthcare practices and fostering collaborative efforts between practitioners, including fostering collaborations between traditional healers and modern healthcare professionals [7,8].

As global interest in alternative and complementary medicine grows [9], understanding the prevalence and scope of traditional healing practices in West Africa becomes paramount for fostering culturally sensitive and effective healthcare interventions. Therefore, gathering and synthesizing information is essential to provide a comprehensive overview of traditional healing practices in West Africa. One way of doing this is to identify key themes, patterns, and trends in the literature related to traditional healing practices, thereby allowing for a nuanced understanding of the prevalence and variations of these practices across different regions, communities, and culture contexts in West Africa. By systematically reviewing the existing literature, gaps in knowledge and areas where further research is needed can be identified, and future studies designed.

This scoping review aimed to explore the landscape of traditional medicine use for health in West Africa, a region known for its diverse ethnicities, languages, and cultures and a rich tapestry of indigenous healing practices [10]. The review systematically outlines the various modalities employed in traditional medicine and discusses the implications of these findings for public health in West Africa.

## Methods

The study design was guided by the Joanna Briggs Institute (JBI) scoping review methodology [11]. The study was reported following the Preferred Reporting Items for Systematic Reviews and Meta-Analyses Extension for Scoping Reviews guidelines (PRISMA-ScR) [12,13].

### Identifying research question

The review was guided by the research question: What is the extent and nature of English-language publications on utilizing traditional medicine for treating various ailments in West Africa?

### Identifying relevant studies

A systematic literature search was conducted in PubMed, Web of Science, and the Cumulative Index of Nursing and Allied Health Literature (CINAHL) using the terms shown in S1 File. The search was conducted from the inception of the database till September 2023. A search of the references of the publications included in the scoping review was also carried out.

### Study selection

Publications identified through the search strategy were downloaded into Endnote and imported into Rayyan. After that, duplicate publications were removed. Three researchers (ORA, MTO, MOF) independently screened the titles and abstracts of the downloaded articles using pre-defined inclusion and exclusion criteria. Studies were included if there was an agreement among the three researchers who performed the screening. Disparities were settled by consensus. In addition, the three researchers completed the full-text review (ORA, MTO, MOF). Uncertainty regarding whether publications met the inclusion criteria was resolved via consensus among the three researchers.

### Inclusion criteria

Peer-reviewed journal articles that covered traditional medicine and health focusing on West Africa. Studies included were published in the English Language, addressed preventive and curative aspects of traditional medicine, and explored herbal and traditional remedies. Studies reporting outcomes related to the impact of traditional medicine on preventive and curative health were included. There was no restriction on study design or date of publication.

### Exclusion criteria

Animal studies were excluded. Also excluded were studies on non-African populations. In addition, unpublished theses and dissertations, letters to the editor, commentaries on studies, scoping, systematic and narrative reviews, and studies whose full lengths could not be accessed were excluded. Also, studies with insufficient results suitable for analysis were excluded.

### Data charting process

A data-charting form was developed to extract relevant variables. The charted variables were the literature characteristics (authors, year of publication, country where study was conducted), study aim, study design, form of traditional medicines used, diseases managed, and the outcomes of the studies.

### Data analysis

The results of the scoping review were reported according to the PRISMA-ScR checklist. A deductive analysis was conducted using the framework developed for the data extraction. Details generated from the analysis were the years of publication, the country in West Africa where studies were conducted, forms of traditional medicines used, diseases (and diseases

pathogenic organisms) investigated, plants (and plant extracts) investigated, and forms and routes of administration of the traditional medicines for the management of diseases.

A descriptive analysis was conducted highlighting the years of publication, countries of publication, study designs, plant families and plant parts used for making traditional medicines, forms in which the traditional remedies are produced, family of plants used for the traditional remedy, and the diseases the traditional remedies are made to prevent or cure. A compilation o of the names of the plant families studied was compiled. The findings were presented in tables.

## Results

The search identified 3484 records, which were downloaded into Endnote and imported into Rayyan. After de-duplication, 3395 records remained. After reviewing titles, abstracts, and screening, 226 articles were eligible for full-text screening. On screening the full articles, 13 articles were excluded either because full articles were not available, the focus of the study was not Africa, or the study was not related to health, leaving 46 articles [14–59]. Three articles were also identified through peer review [60–62], making up to 49 articles for this review. Fig 1 shows the flow diagram of the publication screening process.

As shown in Table 1, the publication period for the 49 manuscripts spanned from 1979 to 2023. The distribution across time revealed four (8.2%) were published between 1979 and 1990 [31–33,35], 20 (40.8%) were published between 1991 and 2000 [14–19,20–23,25,38,41–43,46,49,51,57,59], 13 (26.5%) were from 2001 to 2010 [26,36,37,40,44,47,48,50,52–56], seven (14.3%) papers from 2011 to 2020 [27–30,39,58,62], and five (10.2%) papers from 2021 to 2023 [24,34,45,59,61].

Of the 17 West African countries, only nine have publications on traditional medicines. These countries include Nigeria, which accounts for 25 (51.0%) of the publications [14,16–18,20–26,30,31,33–36,38,41–43,48,49,53,56], Ghana accounting for nine (18.4%) of the publications [27–29,39,40,45,50,52,54], Mali [15,37,47,57] and Togo [19,46,51,59] accounting for four (8.2%) publications each, Benin accounting for three (6.1%) publication [51,60,62], Cote d'Ivoire accounting for two (4.1%) publications [44,55], and Sierra Leone [32], Guinea Bissau [58] and Gabon [61], each having one (2.0%) publication. A publication was produced by Togo and Benin [51].

The studies encompassed various types of study design ranging from experimental studies conducted in laboratories to determine the bioactivities of identified traditional remedies [14,17–19,21–23,25,26,28–31,33–35,38,40–43,44,46,49,51–54,56,57,59]; to ethnobotanical studies which are studies focus on the traditional knowledge, practices, and uses of plants by local communities [15,16,20,24,27,32,36,45,58–62]; ethnopharmacological investigations which are the investigation of traditional knowledge related to the medicinal uses of plants, animals, and other natural substances within specific cultural or ethnic groups [37,39,47,55]; and clinical trials which are systematic investigations conducted on human participants to evaluate the safety, efficacy, and effects of medical treatments, interventions, or therapies [48,50].

Table 2 highlights the characteristics of the 14 ethnobotanical and ethnopharmacological surveys included in the scoping review. The studies were conducted in Nigeria (n = 4), Ghana (n = 3), Mali (n = 2), Benin (N = 2), Gabon (N = 1), Sierra Leone (n = 1) and Guinea-Bissau (N = 1).

A hundred and twenty-three plant families were identified for use as medicinal plants. The Suplemental File 2 provides a compendium of these plant. The most cited plant family was *Euphorbiaceae* [16,24,27,32,39,45,58,60– 62].

Page MJ, McKenzie JE, Bossuyt PM, Boutron I, Hoffmann TC, Mulrow CD, et al. The PRISMA 2020 statement: an updated guideline for reporting systematic reviews. BMJ 2021;372:n71. doi: 10.1136/bmj.n71

Identification of studies from other sources – peer review (N=3):

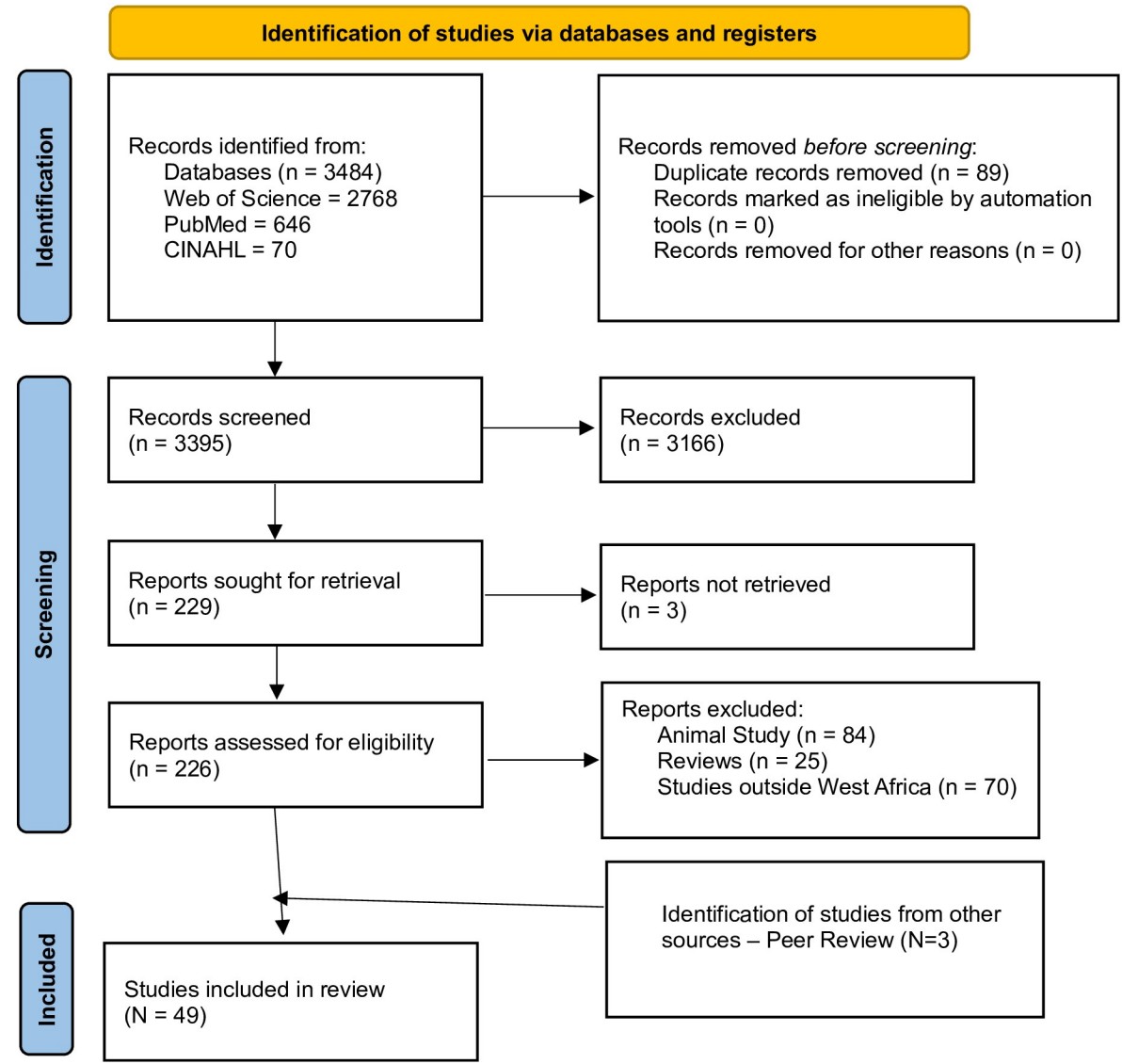

**Fig 1. Flowchart for the review of literature.**

The part of the plants used for medicinal purposes range from the aerial part of plants [58] to barks [15,16,24,27,37,39,45,47,58,60–62], bulbs [60], branches [32,58,60], climbers [39], flowers [39,58], fruits [15,24,27,39,45,47,60,62], gum [15], juice [60], leaves [15,16,24,27,32,36,37,45,47,48,58,60–62], liana [61,62], male plant-inflorescences [16], nuts [60], pods [16,37], rhizome [15,16,39,60,61], roots [15,27,47,24,32,36,37,39,45,58,60–62], sap

**Table 1. Characteristics of published manuscripts on traditional medicines in West Africa.**

| S/no | Title | Author (Year of publication) | Country | Study Aim | Study design |
|---|---|---|---|---|---|
| 1 | Anti-Sickling Potential of *Terminalia catappa* Leaf Extract | Mgbemene et al., 1999 [14] | Nigeria | To investigate the potential of *Terminalia catappa* leaves in managing sickle cell disorders by inhibiting hemolysis and preventing sickling of erythrocytes. | Experimental study |
| 2 | An Ethnobotanical Survey of Herbal Drugs of Gourma District, Mali | Diallo et al., 1999 [15] | Mali | To analyze the knowledge and use of medicinal plants in the Gourma region of Mali and identify potential sources for new bioactive compounds in Malian medicine. | Ethnobotanical survey |
| 3 | Herbs for mental disorders | Nwosu, 1999 [16] | Nigeria | To report on the traditional use of herbal remedies in southern Nigeria for the management of mental disorders. | Ethnobotanical survey |
| 4 | Antidiarrhoeal Activities of *Ocimum gratissimum* (Lamiaceae) | Ilori. 1996 [17] | Nigeria | To investigate the antidiarrhoeal activities of leaf extracts of *Ocimum gratissimum*. | Experimental study |
| 5 | Antimicrobial activities of crude leaf extracts of wilkesiana Acalypha | Alade, 1993 [18] | Nigeria | To investigate the antimicrobial activities of crude leaf extracts of *Acalypha wilkesiana*. | Experimental study |
| 6 | Effects of Three Compounds Extracted from *Morinda lucida* on *Plasmodium falciparum* | Koumaglo, 1992 [19] | Togo | To investigate the effects of three compounds extracted from *Morinda lucida* on the growth of *Plasmodium falciparum* in vitro. | Experimental Study |
| 7 | Antifungal activities of crude extracts of *Mitracarpus villosus* (Rubiaceae) | Irobi, 1993 [20] | Nigeria | To investigate the antifungal activities of crude extracts from *Mitracarpus villosus*. | Experimental study |
| 8 | Effects of crude leaf extracts of *Acalypha torta* against some anaerobic bacteria | Irobi, 1994 [21] | Nigeria | To analyze the antibacterial activities of crude leaf extracts of *Acalypha torta* against certain anaerobic bacteria. | Experimental study |
| 9 | Anti-malarial activity of Nigerian neem leaves | Udeinya, 1993 [22] | Nigeria | To investigate the anti-malarial activity of neem leaf extract. | Experimental study |
| 10 | Antibacterial activity of some medicinal plants from Nigeria | Olukoya, 1993 [23] | Nigeria | To investigate the antibacterial activity of ten medicinal plants used to treat various microbial infections in Nigeria. | Experimental study |
| 11 | Ethnobotanical use-pattern for indigenous fruits and vegetables among selected communities in Ondo State, Nigeria | Olowo, 2022 [24] | Nigeria | To document the ethnobotanical use-pattern of Indigenous fruits and vegetables in selected Ondo State, Nigeria communities. | Ethnobotanical survey |
| 12 | Activities of *Chromolaena odorata* (Compositae) leaf extract against *Pseudomonas aeruginosa* and *Streptococcus faecalis* | Irobi, 1992 [25] | Nigeria | To investigate the antibacterial activities of *Chromolaena odorata* leaf extract against *Pseudomonas aeruginosa* and *Streptococcus faecalis* | Experimental study |
| 13 | Screening of the leaves of three Nigerian medicinal plants for antibacterial activity | Adomi, 2008 [26] | Nigeria | To screen the leaves of three Nigerian medicinal plants for antibacterial activity. | Experimental Study |
| 14 | Documentation of Herbal Medicines Used for the Treatment and Management of Human Diseases by Some Communities in Southern Ghana | Boadu and Asase, 2017 [27] | Ghana | To document herbal medicines traditional healers use to treat and manage human diseases and ailments in southern Ghana. | Ethnobotanical Study |
| 15 | Activity of Herbal Medicines on *Plasmodium falciparum* Gametocytes: Implications for Malaria Transmission in Ghana | Amoah et al, 2015 [28] | Ghana | To assess the efficacy and gametocydal effects of commonly used herbal malaria products in Ghana and their implications for malaria transmission. | Experimental Study |
| 16 | In vitro Antiplasmodial Activities of Aqueous Extracts of Selected Ghanaian Herbal Plants on the Development of Asexual and Sexual Stage Malaria Parasites | Cudjoe et al, 2020 [29] | Ghana | To examine the in vitro antiplasmodial activities of aqueous extracts of selected Ghanaian herbal plants on developing asexual and sexual stage malaria parasites. | Experimental study |
| 17 | In vitro assessment of aqueous and ethanolic extracts of some Nigerian chewing sticks on bacteria associated with dental infections | David et al, 2010 [30] | Nigeria | To investigate the antimicrobial efficacy of aqueous and ethanolic extracts of ten Nigerian chewing sticks on bacteria associated with dental infections. | Experimental study |

(*Continued*)

**Table 1.** (Continued)

| S/no | Title | Author (Year of publication) | Country | Study Aim | Study design |
|---|---|---|---|---|---|
| 18 | Traditional medicine and lead-containing preparations in Nigeria | Healy et al, 1984 [31] | Nigeria | To examine the use of heavy metals, especially lead, in traditional medicines and cosmetics employed by the Nigerian communities in Britain. | Experimental Study |
| 19 | Medicinal plants in Pujehun District of Sierra Leone | MacFoy and Sama, 1983 [32] | Sierra Leone | To document the medicinal plants used by the local people in Pujehun District for the treatment of various diseases and ailments. | Ethnobotanical survey |
| 20 | Antimicrobial Alkaloids from a Nigerian Chewing Stick (*Fagara zanthoxyloides*) | Odebiyi and Sofowora, 1979 [33] | Nigeria | To investigate the antimicrobial properties of alkaloids extracted from a Nigerian chewing stick, *Fagara zanthoxyloide* | Experimental Study |
| 21 | In vitro antiviral activity of twenty-seven medicinal plant extracts from Southwest Nigeria against three serotypes of echoviruses | Ogbole et al, 2021 [34] | Nigeria | To evaluate the antiviral activity of 27 medicinal plant extracts against three serotypes of echoviruses (E6, E7, and E11) that cause various human diseases. | Experimental study |
| 22 | The effect of crude extracts of nine African chewing sticks on oral anaerobes | Rotimi and Mosadomi, 1986 [35] | Nigeria | To test the in vitro susceptibility of four species of oral anaerobic bacteria to crude extracts from nine popular chewing sticks used for dental and oral hygiene in Nigeria. | Experimental study |
| 23 | Use of medicinal plants for the treatment of measles in Nigeria | Sonibare et al, 2009 [36] | Nigeria | To conduct an ethnobotanical survey of three Local Government areas of the Ijebu area of Ogun State in southwest Nigeria for plants used to treat measles. | Ethnobotanical survey |
| 24 | Ethnopharmacological Survey of Six Medicinal Plants from Mali, West-Africa | Grønhaug et al., 2008 [37] | Mali | To collect information about the use of six medicinal plants in the regions around Siby and Dioila, Mali | Ethnopharmacological Survey |
| 25 | Antimicrobial Constituents of the Leaves of *Acalypha wilkesiana* and *Acalypha hispida* | Adesina et al., 2000 [38] | Nigeria | To isolate and identify the antimicrobial principles of *Acalypha wilkesiana* and *Acalypha hispida*. | Experimental study |
| 26 | An ethnopharmacological survey of medicinal plants traditionally used for cancer treatment in the Ashanti region, Ghana. | Agyare et al., 2018 [39] | Ghana | To find out how traditional healers in the Ashanti region of Ghana recognize and classify cancer, how they treat the disease with medicinal plants especially in children, and to investigate the collection, identification, preparation, and application of these plants for cancer treatment. | Ethnopharmacological survey |
| 27 | Herbal Remedies Traditionally Used Against Malaria in Ghana: Bioassay-Guided Fractionation of *Microglossa pyrifolia (Asteraceae)* | Köhler et al., 2002 [40] | Ghana | To investigate the antiplasmodial activity of 11 plants from Ghana, where malaria causes many deaths per year, particularly in young children, and to choose 1 of them, *Microglossa pyrifolia*, for further investigation. | Experimental study |
| 28 | Antibacterial activity of *Alchornea cordifolia* stem bark | Ajali, 2000 [41] | Nigeria | To report the preliminary results of *antimicrobial screening* of the extracts of *Alchornea cordifolia* stem bark | Experimental study |
| 29 | The Antimicrobial Activity of Roots of *Jatropha podagrica* (Hook) | Aiyelaagbe et al., 2000 [42] | Nigeria | To explore the possible antimicrobial agents in *Jatropha podagrica's* roots and identify the active compounds. | Experimental study |
| 30 | Antibacterial activity of *Piliostigma thonningii* stem bark | Akinpelu & Obuotor. 2000 [43] | Nigeria | To determine the antibacterial activity of *Piliostigma thonningii* stem bark extract. | Experimental study |
| 31 | Traditional medicine in North Côte-d'Ivoire: screening of 50 medicinal plants for antibacterial activity | Koné et al., 2004 [44] | Côte-d'Ivoire | To screen 50 medicinal plants used in North Côte d'Ivoire as traditional remedies for bacterial diseases for in vitro antibacterial activity against Gram-negative and Gram-positive bacteria. | Experimental study |

(*Continued*)

**Table 1.** (Continued)

| S/no | Title | Author (Year of publication) | Country | Study Aim | Study design |
|---|---|---|---|---|---|
| 32 | Ethnobotanical Survey and Cercaricidal Activity Screening of Medicinal Plants Used for Schistosomiasis Treatment in Atwima-Nwabiagya District, Ashanti Region, Ghana | Asante-Kwatia et al., 2023 [45] | Ghana | To highlight the importance of medicinal plants in treating schistosomiasis in Ghanaian communities and to investigate their anti-schistosomal activity. | Ethnobotanical survey |
| 33 | Further investigations on the antiviral activities of medicinal plants of Togo | Hudson et al., 2000 [46] | Togo | To determine the antiviral activities of 10 Togolese medicinal plants, previously shown to possess activity against herpes simplex virus. | Experimental study |
| 34 | Ethnopharmacological survey of different uses of seven medicinal plants from Mali, (West Africa) in the regions Doila, Kolokani and Siby | Togola et al., 2005 [47] | Mali | To identify different uses of certain medicinal plants in Mali and investigate their uses in traditional medicine. | Ethnopharmacological survey |
| 35 | Clinical Evaluation of Acalypha Ointment in the Treatment of Superficial Fungal Skin Diseases | Oyelami et al., 2003 [48] | Nigeria | To evaluate the efficacy and safety of *Acalypha wilkesiana* ointment in superficial fungal skin diseases. | Open non-comparative clinical trial |
| 36 | Antibiotic activity of *Aspergillus quadrilineatus* extracts isolated from a Nigerian cereal | Irobi et al., 2000 [49] | Nigeria | To determine the time course for the induction of antibiotic activity and the effects of different carbon sources, pH, temperature, and bivalent cations on the antibiotic activity of *Aspergillus quadrilineatus* extracts. | Experimental study |
| 37 | Evaluation of Efficacy and Safety of a Herbal Medicine Used for the Treatment of Malaria | Ankrah et al., 2003 [50] | Ghana | To determine the efficacy and safety of a decoction, AM-1 was used to treat malaria at an herbal clinic in Ghana. | Clinical trial and experimental study |
| 38 | Antifungal activities of seven West African *Combretaceae* used in traditional medicine | Baba-Mousa et al., 1999 [51] | Benin and Togo | To investigate the antifungal properties of 7 species of the *Combretaceae* family that are abundant in West Africa. | Experimental study |
| 39 | Antimicrobial activity of the leaves and flowering tops of *Acanthospermum hispidum*. | Fleischer et al., 2002 [52] | Ghana | To investigate the antimicrobial activity of the leaves and flowering tops of *Acanthospermum hispidum*. | Experimental study |
| 40 | Antimicrobial activity of *Pterocarpus osun* stems | Ebi and Ofoefule, 2003 [53] | Nigeria | To examine the antimicrobial properties of *Pterocarpus osun* stems. | Experimental study |
| 41 | Antimicrobial activity of some medicinal plants from Ghana. | Konning et al., 2003 [54] | Ghana | The publication presents the results of a preliminary antimicrobial screening of the methanol extracts of *Aframomum melegueta*, *Piper guineense*, *Xylopia aethiopica*, *Zingiber officinale*, and medicinal plants of Ghana. | Experimental study |
| 42 | Antiparasitic activities of medicinal plants used in Ivory Coast | Okpekon et al., 2004 [55] | Cote d'Ivoire | To evaluate in vitro crude extracts of these plants in a drug screening against malaria, leishmaniasis, African trypanosomiasis, helminths, and scabies. | Experimental study |
| 43 | The leaf essential oil of *Costus afer* Ker-Grawl from Nigeria | Taiwo and Bolanle, 2003 [56] | Nigeria | The absence of reports in the literature on this plant's leaf essential oil composition prompted this investigation. | Experimental study |
| 44 | Anti-malarial Activity of Four Plants used in Traditional Medicine in Mali. | Traore-Keita et al., 2000 [57] | Mali | To evaluate in vitro the anti-malarial activity and the cytotoxicity to macrophages of 4 plants used in traditional medicine for treating periodic fevers. | Experimental study |
| 45 | Medicinal plants of Guinea-Bissau: Therapeutic applications, ethnic diversity and knowledge transfer | Catarino et al., 2016 [58] | Guinea-Bissau | To prepare a comprehensive documentation of Guinea Bissau's medicinal plants. | Ethnobotanical survey |
| 46 | Investigation of Medicinal Plants of Togo for Antiviral and Antimicrobial Activities | Anani et al., 2000 [59] | Togo | To determine the antiviral and antibiotic activities of 19 medicinal plants of Togo. | Experimental study |
| 47 | Diversity and knowledge of plants used in the treatment of snake bite envenomation in Benin | Dossou et al., 2021 [60] | Benin | To contribute to a better knowledge of medicinal plants used in the treatment of snakebite envenomation in Benin | Ethnobotanical survey |

(*Continued*)

**Table 1.** (Continued)

| S/no | Title | Author (Year of publication) | Country | Study Aim | Study design |
|---|---|---|---|---|---|
| 48 | Ethnobotanical survey and phytochemical screening of anti-snakebite plants used in Bissok District of Gabon | Mengome et al., 2021 [61] | Gabon | To report an ethnobotanical survey and phytochemical compounds obtained in aqueous extracts of medicinal plants used to treat snakebites in Bissok, a District located in northern Gabon. | Ethnobotanical survey |
| 49 | Ethnobotanical survey on plants used in the treatment of candidiasis in traditional markets of southern Benin | Fanou et al 2020 [62] | Benin | To identify the medicinal plant species traditionally used to treat candidiasis in traditional markets of southern Benin | Ethnobotanical survey |

[39,58,60,61,62], seeds [15,16,24,36,39,45,58,60], stems [15,16,24,32,39,58,60–62], thorns [15], tubers [39,60], twigs [16,36,37,60], and whole plants [15,27,39,45,58,61,62].

Fig 2 shows that leaves and roots are the most used for medicinal purposes, followed by barks and stems. Other parts, such as fruits, seeds, and whole plants, also contribute to many plant-based remedies.

The prepared remedies could be applied topically applied [15,36,39,60,61], inhaled [16], taken orally [60,61], used as scarification [60], or intranasally applied [15,39]. It can be prepared as a decoction [15,16,27,32,36,37,39,47,60,62], powder [15,37,39,47,62], poultice [15,32,39,61], oil [39], juice [15,32,36,39,61], macerated [15,39,47,60–62] or taken as food [24,58,60], soup [36,39], tea [39] or taken as raw fruit [39]. The remedies could also be used for bathing [36], as an enema [15], for infusion [16,27,32,47,61], as a fumigant [15], or for gargling [15,32,37], triturate [60,61] and shampoo [15].

As shown in Fig 3, the forms in which the remedies are prepared are decoction [N = 10], macerated [N = 6], power [N = 5], juice [N = 5], poultice [N = 4], taking as food [N = 3], soup [N = 2], tea [N = 1], oil [N = 1], and taken as raw fruits [N = 1]. Forms in which the remedies are administered are gargling, triturate, bathing, enema, fumigant, and shampoo. These remedies can be administered topically, inhaled, orally, intranasally, or using scarification.

The remedies were used to manage general symptoms such as aches and pains [15,24,32,37,47,58], diarrhea [15,24,32,37], odema, wounds, and burns [15,24,32,37,47,58], dizziness [15,24], convulsion [24,27,32], jaundice [15,27,37,47], bites and sting [24,27,37,58,60,61].

The remedies are also applicable to specific diseases like malaria/fever [15,24,27,32,37,47,58], measles [24,32,36], diabetes [24,27,58], candidiasis [62], sexually transmitted diseases [24,32,37,58], yellow fever [24,32,37], leprosy cholera [24,37], sinusitis [15], cancer [24,39] tuberculosis [24], and schistosomiasis [32,45,47].

Organ disorders managed are anemia and blood disorders [24,27,32,58], bone fracture [24], cold, cough and respiratory diseases [15,24,32,37,58], diseases of the eyes [15,24,32,47,58], diseases of the kidney [37,58], diseases of the liver [24,47,58], the gastrointestinal problem [15,24,27,32,37,47,58], skin problems [15,24,27,32,37,47,58], haemorrhoid [15,27,32,47,58], mental and neurological disorders [16,24,58], neuromuscular problems [15,24,27,47,58], gynaecological problems [24,27,32,37,47,58], and cardiovascular problems [24,27,32,47].

Other non-specific disorders are child health problems [27,32,47,58], oral health problems [15,24,27,32,37], and hernia [27,58]. The remedies are also used as anthelmintic [15,24,37], antiseptic [15], diuretic [27], for the management of poisoning [32,58], baldness [27], dropsy [37], foot rot [27], and to prevent frequent urination when drunk [32].

**Table 2. Characteristics of the ethnobotanical and ethnopharmacological surveys included in the scoping review.**

| S/no | Author (Year of publication) | Country | Forms in which traditional medicines are used | Part of the plant used for traditional medicine | Diseases managed by traditional medicines | Family of plants used for making traditional medicines |
|---|---|---|---|---|---|---|
| 1 | Diallo et al., 1999 [14] | Nigeria | Decoction, powder, boiled fruit added to meals, maceration, shampoo, juice, gargling, fumigation, enema, topical application, intranasal application, poultice, | Stems, leaves, fruit, root, root bark, thorns, gum, whole plant, rhizome, seeds | Abdominal pain, jaundice, dizziness, stimulate appetite, malaria, against heart disease, cold, nausea, vomiting, diarrhoea, dysentery, chest pain, inflammations, wound management, sinusitis, dyspepsia, haemorrhoid, constipation, tonsillitis, gum inflammation; conjunctivitis, asthma, expel roundworms, toothache, antiseptic, oedema, spots and pimples, headache, flatulence, pneumomia, boils, sprain, body pain, laxative | Aizoaceae, Arecaceae, Apiaceae, Asclepiadaceae, Balanitaceae, Bombacaceae, Burseraceae, Capparaceae, Celastraceae, Combretaceae, Cucurbetaciae, Cyperaceae, Fabaceae, Malvaceae, Meliaceae, Nymphaeaceae, Rhamnaceae, Poaceae, Rubiaceae, Salvadoraceae, Sterculiaceae, Tiliaceae |
| 2 | Nwosu, 1999 [16] | Nigeria | Decoctions, aqueous extracts, infusions, stem exudate, inhalation | Leave, twig, stem bark, root bark, seed, male plant- inflorescences, ginger rhizomes, pod | Mental disorders including amnesia, insomnia, senile dementia, depression, delusion of persecution, delusion of grandeur, schizophrenia, and psychosomatic disorders | Amaranthaceae, Liliaceae, Nyctaginaceae, Rubiaceae, Fabaceae, Euphorbiaceae, Meliaceae, Menispermaceae, Verbenaceae, Zingiberaceae, Myristicaceae, Bignoniaceae, Passifloraceae, Myrtaceae, Loganiaceae. |
| 3 | Olowu, 2022 [24] | Nigeria | Not stated | Leaves, fruits, bark, seeds, stems, roots, whole plant, stem, | Anaemia, arthritis, asthma, blood tonic, body pains, boils, bone fractures, boosts libido, cancer, cholera, cleanses liver, colds, conjunctivitis, constipation, convulsions, coughs, diabetes, diarrhoea, dizziness, dog bites, dysentery, earache, eczema, immune booster, enhances lactation, epilepsy, fertility, fever, fibroids, gonorrhoea, headache, heart attack, hypertension, improves digestion, increases blood flow, inflammation, laxative, leprosy, madness, malaria, measles, pneumonia, promotes labour, rheumatism, ringworm, scabies, scorpion bites, skin infection, snake bites, sores, toothache, tumours, tuberculosis., typhoid, ulcer, vomiting, wound healing, yellow fever, hepatitis, good eyesight. | Asteraceae, Acanthaceae, Moraceae, Basellaceae, Sapindaceae, Nyctaginaceae, Crassulaceae, Malvaceae, Amaranthaceae, Sapotaceae, Rutaceae. Lamiaceae, Euphorbiaceae, Arecaceae. Fabaceae, Irvingiaceae, Solanaceae, Anacardiaceae, Talinaceae |
| 4 | Boadu and Asase, 2017 [27] | Ghana | Decoctions, infusions. | Leaves, whole plant, root, bark, fruit | High blood pressure, stops bleeding, piles, diabetes, malaria, typhoid fever, convulsion, menstrual disorders, stroke, halitosis, baldness, Heartburns, sexual disorders, osteoarthritis, diuretic, stomach ulcer, low sperm count, Snakebite, infertility, hernia, miscarriage, foot rot, retarded growth, rheumatism, jaundice, Pruritus, blood tonic. | Acanthaceae, Aloaceae, Amaranthaceae, Annonaceae, Apocynaceae, Asclepiadaceae, Asteraceae, Bignoniaceae, Bombacaceae, Boraginaceae, Capparidaceae, Chrysobalanaceae, Combretaceae, Cucurbitaceae, Cyperaceae, Dennstaedtiaceae, Euphorbiaceae, Fabaceae, Lamiaceae, Malvaceae, Meliaceae, Menispermaceae, Moringaceae, Myrtaceae, Rutaceae, Sapindaceae, Sapotaceae, Solanaceae. |

(*Continued*)

**Table 2.** (Continued)

| S/no | Author (Year of publication) | Country | Forms in which traditional medicines are used | Part of the plant used for traditional medicine | Diseases managed by traditional medicines | Family of plants used for making traditional medicines |
|---|---|---|---|---|---|---|
| 5 | MacFoy and Sama, 1983 [32] | Sierra Leone | Decoction, infusion, poultice, juice or gargled | Leaves, roots, branches, stem | Wounds, eases labour pains, stomach pain, reduce swollen stomach, abdominal pains, gonorrhoea, bilharzia, poison of the stomach caused by witches, skin inflammation, yellow fever, laxative, malaria, cough, palpitation of the heart, toothache, gum pain, haemostasis, dysentery, bilharzia, diarrhea, convulsion, closure of the frontal suture in babies, heals inflammation, prevents frequent urination when drunk, eye troubles, haemorrhoids, causes abortion during early pregnancy, vaginal rash, eruptive skin disease, boil, measles, asthma, promotes closure of the frontal suture in babies, controls persistent menstruation. | Amaranthaceae, Anacardiaceae, Anonaceae, Araceae, Asclepiadacae, Caesalpiniaceae, Caricaceae, Celastraceae, Compositae, Convolculaceae, Connaraceae, Crassulaceae, Ditleniaceae, Ebenaceae, Euphorbiaceae, Flacourtiaceae, Hypericaceae, Labiatae, Loganiaceae, Malvaceae, Melastomataceae, Meliaceae, Mimosaceae, Moraceae, Palmae, Papilionaceae, Passifloraceae, Pedaliaceae, Portulaceae, Rhamnaceae, Rubiaceae, Rutaceae, Selaginellaceae, Solanaceae, Zingiberaceae |
| 6 | Sonibare, 2009 [36] | Nigeria | Taken as soup, juice of the fresh leaves, decoctions, ointment, mixed into soap for bathing | Twigs, roots, seeds, leaves | Measles | Acanthaceae, Asclepiadaceae, Asteraceae, Bignoniaceae, Cucurbitaceae, Dioscoreaceae, Labiatae, Leguminosae, Loranthaceae, Palmae, Papaveraceae, Piperaceae, Poaceae, Sapindaceae, Sapotaceae, Solanaceae, Tiliaceae, Zingiberaceae |
| 7 | Grønhaug et al., 2008 [37] | Mali | Decoction or powder Gum is used as a gargle. | Leaves, roots, stem bark, twig, pod | Pain, fever, diarrhoea, blennorrhoe, wound, chest-affections, constipation, leprosy, sore throat, dysentery, dropsy, ascites and oedemas, jaundice, yellow fever, dermatitis, malaria, stomachache, purgative gonorrhoea, kidney stone, dental abscess, headache, anthelmintic, dysmenorrhea, antidote against scorpion sting and snake bite | *Biopyhtum petersianum* (Oxalidaceae), *Cola cordifolia* (Sterculiaceae), *Combretum mole* (Combretaceae), *Opilia celtidifolia* (Opiliaceae), *Parkia biglobosa* (Leguminosae) and *Ximenia americana* (Olacaceae). |
| 8 | Agyare et al., 2018 [39] | Ghana | Poultice, oil, decoction, raw fruit, exudate, maceration (topical), tea, juice massage, soup, squeezed fruit for drink, powder (topical/oral), cream, nasal drop, tincture. | Roots, bark, leaves, fruit, climbers, stems, rhizome, sap, tuber, seed, whole plant, flower. | Prostrate, stomach, breast, brain, lungs, skin, cervical, ovarian, genital, bone, joint, heart, and throat cancer | Acanthaceae, Amaranthaceae, Amaryllidaceae, Anacardiaceae, Annonaceae, Apocynaceae, Araceae, Araliaceae, Arecaceae, Asteraceae, Boraginaceae, Bromeliaceae, Burseraceae, Calophyllaceae, Caricaceae, Casuarinaceae, Celastraceae, Cleomaceae, Clusiaceae, Combretaceae, Connaraceae, Crassulaceae, Crassulaceae, Cucurbitaceae, Cyperaceae, Dioscoreaceae, Euphorbiaceae, Fabaceae, Lamiaceae, Malvaceae, Marantaceae, Meliaceae, Menispermaceae, Moraceae, Musaceae, Myristicaceae, Myrtaceae, Nyctaginaceae, Pandaceae, Papaveraceae, Passifloraceae, Phyllanthaceae, Piperaceae, Poaceae, Polygalaceae, Portulaceae, Rubiaceae, Rutaceae, Salicaceae, Sapindaceae, Solanaceae, Verbenaceae, Zingiberaceae, Zygophyllaceae |

(*Continued*)

**Table 2.** (Continued)

| S/no | Author (Year of publication) | Country | Forms in which traditional medicines are used | Part of the plant used for traditional medicine | Diseases managed by traditional medicines | Family of plants used for making traditional medicines |
|---|---|---|---|---|---|---|
| 9 | Asante-Kwatia et al., 2023 [45] | Ghana | Not stated | Leaves, whole plant, roots, stem bark, seed, fruit | Schistosomiasis | Asteraceae, Euphorbiaceae, Liliaceae, Apocynaceae, Rhizophoraceae, Meliaceae, Zygophyllaceae, Zingiberaceae, Dichapetalaceae, Acanthaceae, Loranthaceae, Cucurbitaceae, Rubiaceae, Myrtaceae, Solanaceae, Olacaceae, Leguminoseae, Annonaceae, Rutaceae |
| 10 | Togola et al., 2005 [47] | Mali | Decoction, infusion of powder, and maceration | Leaves, stem bark, root, fruits. | Malaria, abdominal pain, dermatitis, amenorrhoea, urinary bilharzias, sterility, baby thinness, ocular infections, high blood pressure, hepatitis, jaundice, chest pain, arthritis, headache, wound healing, haemorrhoids, backache, abdominal flatulence, child malnutrition | Opiliaceae, Loganiaceae, Papilionoideae, Boraginaceae, Meliaceae, Leguminosae, Caesalpinioideae, Cochlospermaceae. |
| 11 | Catarino et al., 2016 [58] | Guinea-Bissau | Food | Flowers, stems, roots, whole plants, branches, bark, leaves, stem, seeds, sap, aerial part of plants | Health problems managed using the plants are pains, pregnancy, childbirth, breastfeeding and diseases of the newborn related problems, gastrointestinal problems, skin inflammations, wounds and burns, cough and respiratory diseases, malaria/fever, stings, bites and poisoning, mental and neurological disorders, sexually transmitted diseases, anemia and blood disorders, male impotence, diseases of the eyes, diseases of the liver, diseases of the kidney, rheumatism and arthritis, hemorrhoids, heart conditions, bones and joints, neuromuscular problems, inguinal hernia, diabetes, and also some unspecified health conditions. | Acanthaceae, Amaranthaceae, Anacardiaceae, Annonaceae, Anisophylleaceae, Apocynaceae, Arecaceae, Asteraceae, Asparagaceae, Bignoniaceae, Cannabaceae, Capparaceae, Caricaceae, Celastraceae, Chrysobalanaceae, Combretaceae, Connaraceae, Convolvulaceae, Cucurbitaceae, Cyperaceae, Dilleniaceae, Dioscoreaceae, Ebenaceae, Euphorbiaceae, Fabaceae, Gentianaceae, Hypericaceae, Hypoxidaceae, Icacinaceae, Lamiaceae, Lauraceae, Loganiaceae, Loranthaceae, Malpighiaceae, Malvaceae, Melastomataceae, Meliaceae, Menispermaceae, Moraceae, Moringaceae, Olacaceae, Opiliaceae, Orchidaceae, Papaveraceae, Passifloraceae, Phyllanthaceae, Piperaceae, Plantaginaceae, Poaceae, Polygalaceae, Rubiaceae, Rutaceae, Sapindaceae, Sapotaceae, Simaroubaceae, Smilacaceae, Solanaceae, Thymelaeaceae, Verbenaceae, Vitaceae, Zingiberaceae |

(*Continued*)

**Table 2.** (Continued)

| S/ no | Author (Year of publication) | Country | Forms in which traditional medicines are used | Part of the plant used for traditional medicine | Diseases managed by traditional medicines | Family of plants used for making traditional medicines |
|---|---|---|---|---|---|---|
| 12 | Dossou et al, 2021 [60] | Benin | Mastication, pounding, decoction, maceration, incineration, trituration. These applications range from local to oral and scarification. | Leaves, roots, bark, fruits, nuts, seed, bulb, juice, roots, sap, reproductive organ, stem, tuber, twigs, regime, rhizome. | Snakebites | Alliaceae, Aloaceae, Amaranthaceae, Anacardiaceae, Annonaceae, Apocynaceae, Arecaceae, Asclepiadaceae, Asteraceae, Bignoniaceae, Bombacaceae, Boraginaceae, Burseraceae, Capparaceae, Caryophyllaceae, Celastraceae, Combretaceae, Convolvulaceae, Cucurbitaceae, Cyperaceae, Dioscoreaceae, Ebenaceae, Euphorbiaceae, Hypoxidaceae, Lamiaceae, Leguminosae, Loganiaceae, Malvaceae, Melastomataceae, Meliaceae, Moraceae, Moringaceae, Musaceae, Myrtaceae, Nymphaeaceae, Onagraceae, Pedaliaceae, Poaceae, Polygalaceae, Portulacaceae, Primulaceae, Rutaceae, Sapindaceae, Sapotacea, Smilacaceae, Solanaceae, Sterculiaceae, Verbenaceae, Vitaceae, Zingiberaceae, Zygophyllaceae. |
| 13 | Mengome et al, 2021 [61] | Gabon | Marcerate, poultice, infusion, mastication, oral, topical, triturate | Leaves, leave stem, rhizome, liana, whole plant, sap, roots, barks | Snakebites | Acanthaceae, Anacardiaceae. Apocynaceae, Asteraceae, Burseraceae, Caricaceae, Caryophyllaceae, Clusiaceae, Guttiferae, Euphorbiaceae, Lamiaceae, Loganiaceae, Melastomaceae, Menispermaceae, Passifloraceae, Piperaceae, Rubiaceae, Rutaceae, Verbenaceae, Vitaceae, Zingiberaceae |
| 14 | Fanou et al, 2020 [62] | Benin | Decoction, pounding, powder, marcerate. | Bark, leaves, leaf stem, fruit, root, whole plant, liana, stem, sap | Candidiasis | Amaryllidaceae, Amaranthaceae, Anacardiaceae, Annonaceae, Apocynaceae, Bignoniaceae, Boraginaceae. Burseraceae, Capparaceae, Caricaceae, Clusiaceae, Combretaceae, Compositae, Connaraceae, Cucurbitaceae, Dichapetalaceae, Euphorbiaceae, Hypoxidaceae, Icacinaceae, Iridaceae, Lamiaceae, Leguminosae, Lythraceae, Malvaceae, Marantaceae, Meliaceae, Musaceae, Myrtaceae, Olacaceae, Phyllanthaceae, Piperaceae, Poaceae, Polygonaceae, Rubiaceae, Rutaceae, Salicaceae, Sapindaceae, Solanaceae, Verbenaceae, Zingiberaceae, Zygophyllaceae |

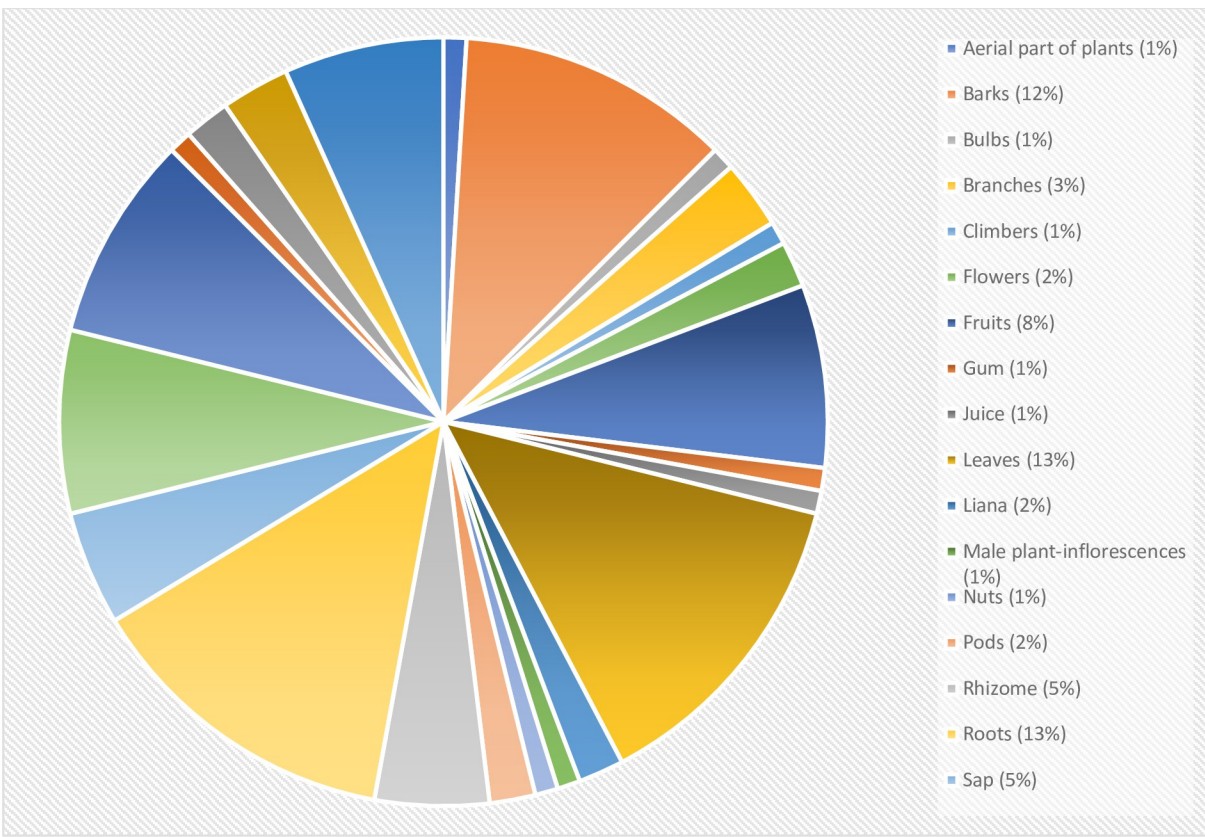

**Fig 2. Piechart representing the parts of the plants used for traditional medicines.**

Table 3 highlights the characteristics of the scoping review's 36 experimental studies and clinical trials. These studies were conducted in Nigeria (n = 22), Ghana (n = 7), Togo (n = 4), Cote d'Ivoire (n = 2), Republic of Benin (n = 1), Mali (n = 1). A single study was conducted in Togo and the Republic of Benin [51].

The effect of medicinal plants was explored against the following group of diseases and disorders: sickle cell [14], gastrointestinal disorders including diarrhea and dysentery [17,23,31,34,54], respiratory disorders like cough [23,34,43,54,56], malaria/fever [19,22,28,29,40,41,50,55–57], dental infection [30,33,35], skin infections [20,23,31,44,54,55] and wound infections [21,43,53]. Other disorders managed were eye infection [31,56], infertility [31], leprosy [41,43], ulcers [43], meningitis/encephalitis [34], smallpox [43], urinary tract infection [44,56], sore throat [23], gonorrheal [23], schistosomiasis [45] and rhematic pain specifically [41,54,56]. Some studies explored specific antifungal [18,20,42,48,49,51], antibacterial [18,20–23,25,26,30,35,42,46,49,52,53,59] and antiviral [34,46,59] effects of plant extracts.

Sixty plant families were studied (See S3 File for the compendium). The most common plant family studied were Euphorbiaceae [18,21,29,38,40,41,42,44,45,47,50,55], Rubiaceae [19,20,26,30,35,40,44,46,55,57,59] and Combretaceae [14,23,30,35,44,51,55]. The 12 studies on Euphorbiacea were conducted in Nigeria [18,21,38,41,42,48], Ghana [29,40,45,50], and Cote D'Ivoire [44,55]. The plant family studied in the most diverse country was the Rubiaceae studied in Togo [19,46,59], Nigeria [20,26,30,35], Ghana [40], Cote D-Ivoire [44,55], and Mali [57]. Combretaceae was studied in Nigeria [14,23,30,35], Cote D'Ivoire [44,55], Togo [51], and the Republic of Benin [51].

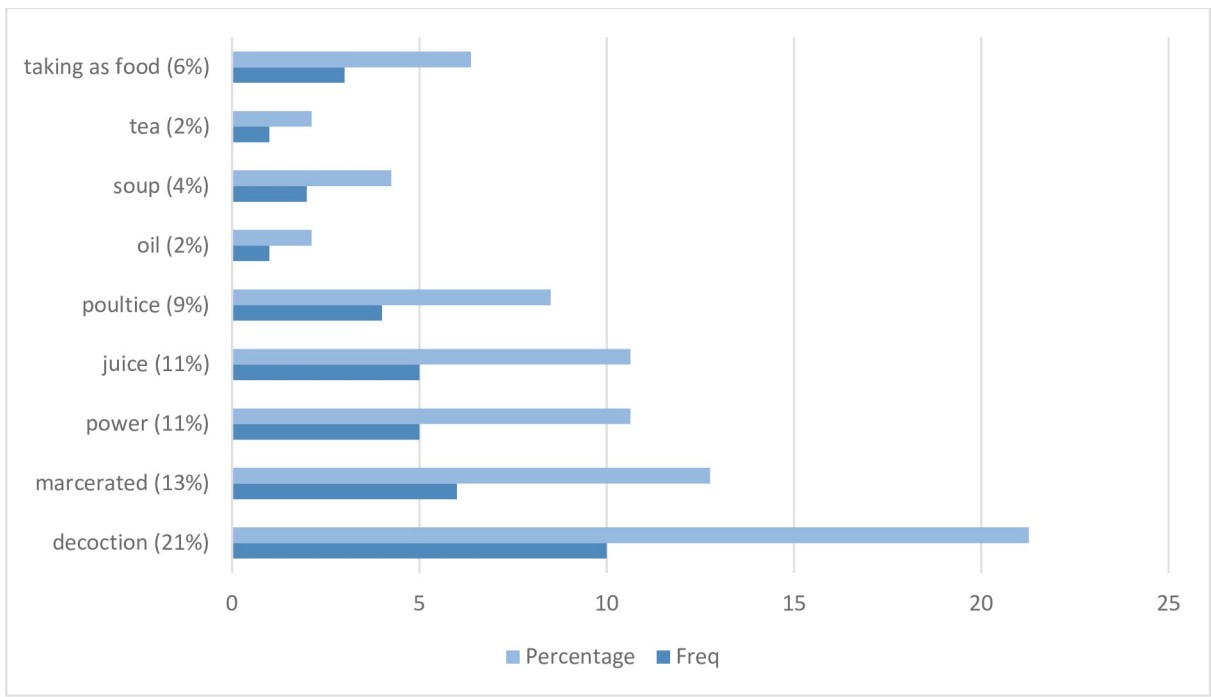

**Fig 3. Histograph representing the forms in which remedies are prepared.**

Plant families reported in studies conducted in Nigeria include Lamiaceae [17], Euphorbiacea [18,21,38,41,41,48], Rubiaceae [20,26,30,35], Mahogany [20], Combretaceae [14,23,30,35], Daisy [25,35], Annonaceae [34], Sapotaceae [30,35], Fabaceae [30,35], Piperaceae [30], Rutaceae [30,33,35], Anacardiaceae [35], Aspergillaceae [49], Leguminosea [53], Costaceae [56], Rutaceae [35] and Madder [35].

Plant families reported in studies conducted in Ghana include Euphorbiacea [29,40,45,50], Daisy [52], Apocynaceae [28,40], Annonaceae [29,35], Moringaceae [29], Anacardiaceae [29], Zingiberaceae [54], Amaranthaceae [40] Asclepiadaceae [40], Asteraceae [40], Bombacaceae [40], Mimosaceae [40], Rubiaceae [40], Mallows [50], Solanaceae [45,50], Legumes [50], Olacaceae [45], Acanthaceae [45], Zingiberaceae [45], Leguminoceae [45], Rhizophoraceae [45], Rutaceae [45], and Loranthaceae [45].

Plant families reported in studies conducted in Togo include Rubiaceae [19,46], Combretaceae [51], Malvacae [46,59], Astraceae [46,59], Commelinaceae [46,59], Bombacaceae [46,59], Davalliaceae [46,59], Moraceae [46,59], Rutaceae [46,59], Simarubaceae [46,59], Sapindaceae [46,59], Apocynaceae [59], Bignoniaceae [59], Orchidaceae [59], Rubiaceae [59], Verbenaceae [59].

Plant families reported in studies conducted in Cote d'Ivoire include Combretaceae [44,55], Annonnaceae [44,55], and Lythraceae [55]. Agavaceae [44], Amaranthaceae [44], Anacardiaceae [44], Apocynaceae [44,55], Araliaceae [44], Asparagaceae [44], Bignoniaceae [44], Caesalpinaceae [44], Celastraceae [44], Chrysobalanaceae [44], Cochlospermaceae [44], Convolvulaceae [44], Cyperaceae [44], Euphorbiaceae [44,55], Fabaceae [44], Hymenocardiaceae [44], Hyppocrateaceae [44], Loganiaceae [44], Malvaceae [44], Meliaceae [44,55], Mimosaceae [44], Moraceae [44], Olacaceae [44], Opiliaceae [44,55], Polygalaceae [44], Rubiaceae [44,55], Sterculiaceae [44], Verbenaceae [44,55], Vitaceae [44], Combretaceae [55], Papilionaceae [55], Passifloraceae [55], Sapindaceae [55], and Zingiberaceae [55].

**Table 3. Characteristics of the experimental studies and clinical trials included in the scoping review.**

| S/no | Author (Year of publication) | Country | Family of plant used for traditional medicine | Diseases studied | Study findings |
|---|---|---|---|---|---|
| 1 | Mgbemene et al., 1999 [14] | Nigeria | *Terminalia catappa* leaf ethanol extract (Combretaceae) | Sickle cell disorder | The extract inhibited osmotically induced hemolysis of human erythrocytes in a dose-dependent manner. It also prolonged the clotting time of uncoagulated blood. Additionally, a 1.0 mg/ml solution of the extract effectively prevented and reversed the sickling of human 'SS' erythrocytes induced by a sodium metabisulphite solution. |
| 2 | Ilori, 1996 [17] | Nigeria | Extracts of *Ocimum gratissimum* leaf (Lamiaceae) | Diarrhoeal diseases | The extracts were active against *Aeromonas sobria*, *Escherichia coli*, *Plesiomonas shigelloses*, *Salmonella typhi*, and *Shigella dysenteriae*, but most active against *Shigella dysenteriae* and least active against *Salmonella typhi*. |
| 3 | Alade, 1993 [18] | Nigeria | Water and ethanol extracts of leaves of *Acalypha wilkesiana* (Euphorbiaceae) | Antimicrobial activities of extracts against bacteria and fungi. | The extracts inhibited the growth of bacteria and fungi. The aqueous extract was static, while the ethanolic extract was uniformly tidal. |
| 4 | Koumaglo, 1998 [19] | Togo | Decoctions of the aerial part, stem bark, or root bark of *Morinda lucida* (Rubiaceae) | Malaria and other tropical diseases | The three compounds extracted showed dose-dependent inhibition of the growth of *Plasmodium falciparum* in vitro. |
| 5 | Irobi, 1993 [20] | Nigeria | Ethanolic extracts of *Mitracarpus villosus* (Rubiaceae) | Skin infections caused by *Trichophyton rubrum*, *Microsporum gypseum*, *Candida albicans*, *Aspergillus niger*, and *Fusarium solani* | Extracts showed definite antifungal activities against the tested fungi. Aqueous extracts and the glycerol vehicle control did not inhibit any of the fungi. The inhibition zones produced by the ethanol extracts ranged from 10 to 20.5 mm. The minimum inhibitory concentration of the extracts ranged from 0.50 to 4.0 mg/ml, and their minimum fungicidal concentration values ranged from 1 to 8 mg/ml. The extracts were fungistatic at lower concentrations and fungicidal at higher concentrations. |
| 6 | Irobi, 1994 [21] | Nigeria | *Acalypha torta* extracts (Euphorbiaceae) | Anaerobic bacteria associated with wound infections. | The aqueous, ethanol, and methanol extracts of *A. torta* showed in vitro antibacterial activities against all the anaerobic bacteria tested. The extracts exerted bactericidal effects on the bacteria at larger doses. |
| 7 | Udeinya, 1993 [22] | Nigeria | Neem leaves extract (Mahogany) | Malaria | Neem leaf extract exhibited significant anti-malarial activity, leading to the complete cessation of parasite development. |
| 8 | Olukoya, 1993 [23] | Nigeria | Water and ethanol extracts from species of Combretaceae | Gonorrhea, sore throat, diarrhea, abscess, boils, dysentery, coughs, body sores | Several of the tested plants exhibited antimicrobial activity against a range of microorganisms, including bacteria such as Streptococcus, *Neisseria gonorrhoeae*, *Salmonella*, *Shigella*, *Escherichia coli*, and more |
| 9 | Irobi, 1992 [25] | Nigeria | Ethanolic leaf extract of *Chromolaena odorata* (Daisy) | *Pseudomonas aeruginosa* and *Streptococcus faecalis*. | The ethanolic leaf extract of *Chromolaena odorata* exhibited antibacterial activity against both *Pseudomonas aeruginosa* and *Streptococcus faecalis*. The extract was more effective against *Streptococcus faecalis* than *Pseudomonas aeruginosa*. |

(*Continued*)

**Table 3.** (Continued)

| S/no | Author (Year of publication) | Country | Family of plant used for traditional medicine | Diseases studied | Study findings |
|---|---|---|---|---|---|
| 10 | Adomi, 2008 [26] | Nigeria | Aqueous and ethanol extracts of the leaves of *Morinda lucida* (Rubiaceae) | Antibacterial activity | *Morinda lucida* extract was active against all tested bacteria, while the latex of *Alstonia boonei* showed no activity against any of the bacteria tested. |
| 11 | Amoah et al, 2015 [28] | Ghana | Aqueous and ethanol extracts of *Cryptolepis sanguinolenta* (Apocynaceae) and *Azadirachta indica* (Meliaceae) | Malaria | Some herbal anti-malarial products showed potential gametocydal activities, which could reduce the transmission of malaria parasites. |
| 12 | Cudjoe et al, 2020 [29] | Ghana | Extract from *Alchornea cordifolia* (Euphorbiaceae), *Polyalthia longifolia* (Annonaceae), *Moringa oleifera* (Moringaceae), *and Mangifera indica* (Anacardiaceae) | Malaria | *Alchornea cordifolia* exhibited activity against the three tested parasites and effectively cleared both asexual parasites and gametocytes. *Moringa oleifera* and *Mangifera indica* also showed anti-malarial activities. The study also identified the presence of phenolic flavonoids and other phytochemicals in the herbal extracts, which have been associated with anti-malarial activity. |
| 13 | David, 2010 [30] | Nigeria | Chewing sticks made from Sapotaceae, Fabaceae, Combretaceae, Piperaceae, Rutaceae, Meliaceae, Clusiaceae, and Rubiaceae | Bacteria associated with dental infections | The ethanolic extracts had higher antimicrobial activity compared to the aqueous extracts. *Viellaria paradoxical* showed a zone of inhibition ranging between 5 and 13 mm, with *Moraxella catarrhalis* being the most susceptible bacterium to the extracts. *Actinomyces vercosus* was resistant to the extracts of *Piper guineense*, *Azadirachta indica*, and *Massularia accumunata*. |
| 14 | Healy et al, 1984 [31] | Nigeria | Tiro | Eye cleaner | The samples of tiro collected from Babalawo (native doctor), Eleweomo (herbalist), and homemade products in both two and villages contained an average of 50.1% (w/w) of lead. |
| 15 | Odebiyi and Sofowora, 1979 [33] | Nigeria | Ethanolic extract of the root of *Fagara zanthoxyloides* (Rutaceae) | Toothaches | The extract yielded four pure compounds that showed antimicrobial activity. One of these compounds is a tertiary phenolic alkaloid (carithine–6–one), and two are quaternary alkaloids (chelerythrine chloride and berberine chloride), while the structure of the fourth compound was not determined. |
| 16 | Ogbole et al, 2021 [34] | Nigeria | Methanol extracts of dried Annonaceae | Echovirus infections causing meningitis, encephalitis, respiratory illness, gastroenteritis | Out of the 27 extracts tested, 11 showed antiviral activity against at least one echovirus serotype. The extracts had low cytotoxicity, indicating their safety for use. |
| 17 | Rotimi and Mosadomi, 1986 [35] | Nigeria | Crude aqueous or alcoholic extracts from the stems and or roots of *Serindeia warneckei* (Anacardiaceae), *Fagara zanthoxyloides* (Rutaceae), *Distemonanthus benthamianus* (Fabaceae), *Massularia accuminata* (Rubiaceae), *Anogeissus leiocarpus* (Combretaceae), *Vernonia amygdalina* (Daisy), *Bytyrospermum paradoxum* (Sapotaceae), *Terminalia glaucescens* (Combretaceae), *Nauclea latifolia* (Madder) | Dental caries, periodontal disease, and other oral diseases | All the extracts showed inhibitory effects on the growth of the oral anaerobes, with varying degrees of potency. *Serindeia warneckei* had the greatest and most consistent inhibitory effect on the four species studied, namely *Bacteroides oralis*, *Bacteroides melaninogenicus*, *Bacteroides gingivalis*, *Bacteroides asaccharolyticus* |

(*Continued*)

**Table 3.** (Continued)

| S/no | Author (Year of publication) | Country | Family of plant used for traditional medicine | Diseases studied | Study findings |
|------|------------------------------|---------|-----------------------------------------------|------------------|----------------|
| 18 | Adesina et al., 2000 [38] | Nigeria | Aqueous ethanol extract of *Acalypha wilkesiana* and *Acalypha hispida* leaves (Euphorbiaceae) | Antimicrobial activity of the leaves. | Gallic acid, corilagin, and geraniin were identified as the compounds responsible for the observed antimicrobial activity in *Acalypha wilkesiana* and *Acalypha hispida* leaves. |
| 19 | Köhler et al., 2002 [40] | Ghana | Aqueous leaf decoction, leaf extract, and infusion of *Gomphrena celosioides* (Amaranthaceae), *Picralima nitida* (Apocynaceae), *Pergularia daemia* (Asclepiadaceae), *Emilia sonchifolia* (Asteraceae), *Microglossa pyrifolia* (Asteraceae), *Adansonia digitata* (Bombacaceae), *Euphorbia hirta*, *Phyllanthus niruroides* (Euphorbiaceae), *Tetrapleura tetraptera* (Mimosaceae), *Mitragyna inermis*, *Mitragyna stipulosa* (Rubiaceae) | Fever and malaria | Different extracts from the plants showed promising in vitro activity against *Plasmodium falciparum*. |
| 20 | Ajali, 2000 [41] | Nigeria | Stem bark of *Alchornea cordifolia* (Euphorbiaceae) | Fever, rheumatic pains, purgative, and leprosy | *Alchornea cordifolia* stem bark extracts exhibited antimicrobial activities, supporting some uses of the plant in traditional medicine. |
| 21 | Aiyelaagbe et al., 2000 [42] | Nigeria | Stem and roots of *Jatropha podagrica* as chewing sticks (Euphorbiaceae) | Bacterial and fungal infections, including *S. Aureus*, *B. Cereus*, and *Candida albicans* | All the extracts exhibited broad-spectrum antibacterial activity, with the hexane extract of the yellow rootbark being the most active. It showed moderate antifungal activity against *Candida albicans* supporting its traditional use in treating different infections. |
| 22 | Akinpelu and Obuotor. 2000 [43] | Nigeria | Methanolic extract of the stem bark *Piliostigma* thonningii (Fabaceae) | Leprosy, smallpox, coughs, wounds, and ulcers | The extract *of Piliostigma thonningii* stem bark exhibited antibacterial activity against six out of eight tested bacteria at a 20 mg/ml concentration. |
| 23 | Koné et al., 2004 [44] | Côte-d'Ivoire | 67 crude ethanol extracts from leaves, stem barks, roots and whole plants of 50 plants belonging to Agavaceae, Amaranthaceae, Anacardiaceae, Annonaceae, Apocynaceae, Araliaceae, Asparagaceae, Bignoniaceae, Caesalpinaceae, Celastraceae, Chrysobalanaceae, Cochlospermaceae, Combretaceae, Convolvulaceae, Cyperaceae, Euphorbiaceae, Fabaceae, Hymenocardiaceae, Hyppocrateaceae, Loganiaceae, Malvaceae, Meliaceae, Mimosaceae, Moraceae, Olacaceae, Opiliaceae, Polygalaceae, Rubiaceae, Sterculiaceae, Verbenaceae and Vitaceae families | Abdominal pain, boils, conjunctivitis, constipation, cough, dermatitis, diarrhoea, dysentery, wounds, eyes infections, fever, gonorrhoea, heart problems, infected eruption intestinal worms, malaria, mouth wounds, respiratory diseases, rheumatism, sore throat, stomachache, toothache, vomiting | 31 of the 67 extracts showed promising antibacterial activity against both Gram negative (*Escherichia coli* and *Pseudomonas aeruginosa*) and Gram positive (*Staphylococcus aureus*, *Enterococcus faecalis*, *Streptococcus pyogenes* and *Bacillus subtilis*) bacteria. 10 extracts derived from 10 plants belonging to 10 families showed considerable antibacterial activity. The families are Fabaceae, Caesalpinaceae, Sterculiaceae, Euphorbiaceae, Olacaceae, Meliaceae, Anacardiaceae, Vitaceae, Rubiaceae, Moraceae. |
| 24 | Asante-Kwatia et al., 2023 [45] | Ghana | *Ximenia americana* (Olacaceae), *Justicia flava* (Acanthaceae), *Jathropha multifida* (Euphorbiaceae), *Bridelia tenufolia* (Zingiberaceae), *Xylia evansii* (Leguminoceae), *Anopyxis klaineana* (Rhizophoraceae), *Withania somnifera* (Solanacea), *Zanthoxylum zanthoxyliodes* (Rutaceae), *Loranthus lecardi* (Loranthaceae) | *Schistosoma mansoni* Cercariae | All extracts exhibited a concentration and time-dependent cercaricidal activity against *Schistosoma mansoni* cercariae. *Brideli tenufolia* and *Zanthoxylum zanthoxyloides* exhibited the least cercaricidal activity among all extracts tested. |

(*Continued*)

**Table 3.** (Continued)

| S/no | Author (Year of publication) | Country | Family of plant used for traditional medicine | Diseases studied | Study findings |
|---|---|---|---|---|---|
| 25 | Hudson et al., 2000 [46] | Togo | Antiviral activities of 10 species of Togolese medicinal plants that possess activity against herpes simplex virus. The species belong to the following family: Asteraceae, Bombacaceae, Commelinaceae, Davalliaceae, Malvacae, Moraceae, Rubiaceae, Rutaceae, Simarubaceae, Sapindaceae | Herpes simplex virus | The dominant activity of the plants was virucidal, although *Adansonia digitata* extracts also appeared to have intracellular antiviral activities. Long-wavelength ultraviolet light enhanced the anti-herpes simplex virus activity of the seven most active extracts. All the extracts contained antiviral photosensitizers, and the rootbark and leaf extracts of *Adansonia digitata* were the most potent. |
| 26 | Oyelami et al., 2003 [48] | Nigeria | *Acalypha wilkesiana* ointment (Euphorbiaceae) | *Tinea pedis*, *Pityriasis versicolor*, and *Candida intetrigo* | Acalypha ointment was very efficacious in treating these superficial fungal skin diseases, with a cure rate of 100% in each condition |
| 27 | Irobi et al., 2000 [49] | Nigeria | *Aspergillus quadrilineatus* (Aspergillaceae) extracts isolated from a Nigerian cereal | Antibiotic activity of extracts on fungal and bacterial species of clinical importance | *Aspergillus quadrilineatus* extracts demonstrated antibiotic activity against both fungal and bacterial species, indicating their potential as antimicrobial substances |
| 28 | Ankrah et al., 2003 [50] | Ghana | Plant decoction called AM-1 was formulated from *Jatropha curcas* (Euphorbiaceae), *Gossypium hirsutum* (Mallows), *Physalis angulate* (Solanaceae) *and Delonix regia* (Legumes) | Malaria | The AM-1 decoction eliminated malaria parasites (*Plasmodium falciparum* and *Plasmodium malarie*) from the peripheral blood of patients with malaria and did not show any undesired effects in patients or laboratory rats |
| 29 | Baba-Moussa et al., 1999 [51] | Northern regions of Benin and Togo | Antifungal activities of seven species of Combretaceae commonly used in traditional medicine in West Africa | *Candida albicans*, *Epidermophyton floccosum*, *Microsporum gypseum*, *Trichophyton mentagrophytes*, and *Trichophyton rubrum* | *Pteleopsis suberosa* and *Terminalia aicennioides* were the most active plants, and their antifungal activity may be attributed to their high content of tannins and saponins. |
| 30 | Fleischer et al., 2002 [52] | Ghana | The leaves and flowering tops of *Acanthospermum hispidum* (Daisy) | Wide range of pathogenic bacteria | Ethanolic extracts of the leaves and flowering tops of *Acanthospermum hispidum* showed varying degrees of activity against a wide range of pathogenic bacteria, particularly Gram-positive organisms. The activity was mainly observed in the polar fractions of the alcoholic extract, while no activity was observed for the aqueous extract of the fresh plant material. |
| 31 | Ebi and Ofoefule, 2003 [53] | Nigeria | The leaves of *Pterocarpus osun* (Leguminoceae) | Five multidrug-resistant *Escherichia coli* strains obtained from wounds | Methanol, ethanol, and aqueous leaf extracts of *Pterocarpus osun* showed different inhibition zones to the *Escherichia coli* isolates. Extracts of *Pterocarpus* had a broad spectrum of activity on the tested bacteria. |
| 32 | Konning et al., 2003 [54] | Ghana | *Aframomum melegueta* (Zingiberaceae), *Piper guineense* (Piperaceae), *Xylopia aethiopica* (Annonaceae), *and Zingiber officinale* (Zingiberaceae) | Cough, bronchitis, rheumatism, dysenteric conditions, and skin infectious diseases | The methanol extracts of the plants showed significant antimicrobial activity against both Gram-positive and Gram-negative bacteria, as well as fungi. |
| 33 | Okpekon et al., 2004 [55] | Cote d'Ivoire | Extracts from the leaves, roots, stem barks, trunk barks, and whole plants of 17 herbal medicinal plants from 13 families namely Annonaceae, Apocyanaceae, Combretaceae, Euphorbiaceae, Lythraceae, Meliaceae, Opiliaceae, Papilionaceae, Passifloraceae, Rubiaceae, Sapindaceae, Verbenaceae, Zingiberaceae. | Dysentery, helminthiasis, Hypertension, insect bites, jaundice, malaria, scabies, trypanosomiasis | *Anogeissus leiocarpus* and *Terminalia glaucescens*, showed strong activity against *Plasmodium falciparum*. *Lawsonia inermis* was selectively prescribed against trypanosomiasis and showed interesting trypanocidal activities Anthelmintic activities were found for 10 active species, and *Uvaria afzelli* and *Monodora myristica* were active against mites. |

(*Continued*)

**Table 3.** (Continued)

| S/ no | Author (Year of publication) | Country | Family of plant used for traditional medicine | Diseases studied | Study findings |
|---|---|---|---|---|---|
| 34 | Taiwo and Bolanle, 2003 [56] | Nigeria | The leaf essential oil of *Costus afer* (Costaceae) | Rheumatism, eye infections, headache, cough, diuretic, urethral discharge, and malaria | The essential oil of *Costus afer* from Nigeria contains 27 compounds, with sesquilavandulyl acetate being the principal component. However, the essential oil showed no antimicrobial activity |
| 35 | Traore-Keita et al., 2000 [57] | Mali | Aqueous, hydromethanol, and chloroform extractions of *Mitragyna inermis* (Rubiaceae) | Periodic fevers and malaria | The alkaloids in chloroform extracts and ursolic acid purified from the hydromethanol extract showed significant anti-malarial activity, while the aqueous extracts did not show high anti-malarial activity. |
| 36 | Anani et al., 2000 [59] | Togo | Methanol extracts of the leaf, root, bark, or whole plant of 19 medicinal plants used in Togo belonging to the following family: Apocynaceae, Asteraceae, Bignoniaceae, Bombacaceae, Commelinaceae, Davalliaceae, Malvaceae, Moraceae, Orchidaceae, Rubiaceae, Rutaceae, Sapindaceae, Simarubaceae, Verbenaceae | *Staphylococcus aureus, Streptococcus fecalis, Bacillus subtilis, herpes simplex, Escherichia coli, Klebsiella pneumoniae, Salmonella typhimurium, Sindbis, Pseudomonas aeruginosa, poliovirus, Mycobacterium phlei, Candida albicans,* | Extracts showed significant antiviral activity, and all but two displayed antibiotic activity. Three species from the plant families Malvaceae, Asteraceae, Commelinaceae, were active against all three test viruses. Seven plants showed activity against only one or two viruses. The antibiotic profiles varied considerably, suggesting the presence of different bioactive phytochemicals in each species. |

The plant family reported in Benin was Combretacea [51]. The plant family studied in Mali was Rubiaceae [57].

## Discussion

This scoping review maps herbal remedies used for medicinal purposes in humans in West Africa. The findings indicate that the study of traditional medicines has continued and increased over time, thereby underscoring the enduring relevance of traditional medicine research, with contributions consistently observed over the years. There has also been an extensive exploration of medicinal plants across West African countries, including their use for managing various diseases and disorders. The geographic distribution of these studies emphasizes the regional relevance of traditional medicine in addressing diverse health challenges. However, the number of countries in West Africa publishing experimentations on traditional remedies, especially clinical trials and ethnobotanical and ethnopharmacological surveys of plants used for traditional medicine, is low, with research skewed towards Nigeria and Ghana.

One of the strengths of this study lies in its thorough examination of research about traditional medicine used in managing diseases in West Africa—a continent widely recognized for its considerable reliance on traditional healthcare practices for the well-being of its population [63]. However, the study had some limitations. First is the inclusion of only publications available and accessible in English, and second is the search limit to three databases employed. These may have inadvertently excluded some research publications on traditional medicines used in West Africa since researchers in the sub-region also publish in French and other languages. Despite this limitation, the present study provides valuable insights into the subject matter.

First, diverse study designs have been employed to investigate traditional medicine in the region. The studies were majorly experimental in design, focusing on laboratory-based investigation of identified traditional remedies. In addition, a few clinical trials, representing systematic investigations on human participants, were conducted to evaluate the safety and efficacy

of some traditional medical treatments. There is, however, the need for a lot more experimental and clinical trial studies that can facilitate the translation of the traditional medicines found to have therapeutic effects into commercial products, as the validation of indigenous drugs is relevant to modern societies at large and helps to sustain local health care practices [64].

Furthermore, ethnobotanical and ethnopharmacological studies provided insights into plants' traditional knowledge, practices, and uses, contributing valuable cultural context to the exploration. In addition, the ethnobotanical and ethnopharmacological surveys provide a deeper understanding of the traditional knowledge embedded in local communities. The surveys conducted in West Africa spotlighted a diverse array of plant families, plant parts, and preparation methods utilized in traditional remedies. Despite this wealth of knowledge, there remains untapped potential for further exploration using research methods tailored to the practice of traditional medicines [65].

These insights can be enhanced through dedicated studies focusing on specific diseases and their traditional medical remedies, as undertaken by those publications on schistosomiasis [45], cancer [39], sickle cell [14], malaria [22,29,40,50], snake bites [60,61], candidiasis [62], and Herpes simplex virus [46]. Conducting disease-centric studies has the potential to aid the sub-region in developing therapeutic measures for disease entities categorized as neglected tropical diseases [66], which are of interest to the sub-region and currently receive limited global attention and funding. Such disease-specific studies on herbal remedies are growing. These include reviews on herbal therapies for diarrhoea in sub-Saharan Africa [67] and the global use of traditional medicines for cancer [68]. These disease-specific studies generate evidence for evaluating the efficacy and safety of these herbal remedies, which is crucial for integrating traditional medicine into mainstream healthcare systems to help inform medical decision-making based on empirical data. Further studies are needed to compare the disease-specific herbal therapies used between regions and their efficacy and safety profile.

The current study highlights the diverse therapeutic applications of identified traditional remedies, encompassing the management of symptomatic conditions and the targeted treatment of specific diseases. However, the efficacies of these therapies remain inadequately studied, with only two identified clinical trials [48,50]. To address this gap, countries should actively promote research on traditional medicines by formulating policies and guidelines and funding such endeavors. Developing policies and guidelines that facilitate the seamless integration of traditional medicines into mainstream healthcare can be crucial in encouraging and supporting clinical trials in this field.

Nigeria [69,70] and Ghana [71] are at the forefront of these regional integration efforts. These pioneering initiatives aim to bridge the gap between traditional and modern medical practices and foster a more comprehensive approach to healthcare that embraces the rich heritage of traditional healing methods. The progress made by these nations in the number of publications on traditional medicines reflects a dedicated commitment to exploring the potential benefits of traditional medicine within the broader healthcare framework. As these integration efforts unfold, they promise to expand therapeutic options for patients, foster collaborative healthcare practices, and preserve the invaluable traditional knowledge passed down through generations. There are also ongoing efforts in Sierra Leone, led by the World Health Organisation, to regulate the practice of traditional medicines in the country. The observed regional disparity in the countries contributing information on using plants for traditional medicines emphasizes the need for broader engagement and collaboration to enrich the diversity of perspectives and practices captured in research.

Frequently referenced plant families in medicinal studies in West Africa include Euphorbiaceae, Rubiaceae, Combretaceae, Meliaceae, and Euphorbiaceae, has gained global recognition for their effectiveness in treating gastrointestinal disorders, respiratory complaints, skin

problems, and inflammation and injuries [72]. Within the *Rubiaceae* family, plant species predominantly found in warmer climates are indicated for treating diverse conditions such as malaria, hepatitis, eczema, edema, cough, hypertension, diabetes, and sexual dysfunction. Biological screenings of plants used by traditional healers have demonstrated various beneficial activities, including anti-malarial, antimicrobial, antihypertensive, antidiabetic, antioxidant, and anti-inflammatory properties [73].

Combretaceae species find extensive use in traditional medicine to address inflammation, infections, diabetes, malaria, bleeding, diarrhea, and digestive disorders and are employed as diuretics [74]. In the Meliaceae family, while various species are utilized for purposes such as vegetable oil, soap-making, and insecticides, *Trichilia emetica*, a plant native to Africa, holds significance in traditional medicine for treating a range of ailments such as abdominal pains, dermatitis, hemorrhoids, jaundice, and chest pain [75].

As indicated in the current study, the medicinal properties of plants can originate from various plant components such as leaves, roots, bark, fruits, seeds, and flowers. Each plant part may harbor distinct active ingredients, contributing to its therapeutic effects. We found that leaves and roots are the region's most used parts for medicinal purposes. Leaves, often rich in bioactive compounds, are frequently utilized for their therapeutic properties in various remedies and are prepared in various forms ranging from decoctions to poultices. Roots are also valued for their medicinal properties and are often prepared as decoctions or infusions to extract their active ingredients. The range of plant parts used for medicinal purposes and the forms used in the current study reflect the diverse traditional knowledge and practices embedded in local communities across West Africa.

This scoping review, however, highlights the need for broader regional engagement and collaboration on traditional medicines' use to capture diverse perspectives and practices in West Africa, such as conducting more experimental and clinical trial studies to validate Indigenous medicine's use and facilitate their translation into commercial products. This validation is crucial for local healthcare practices, improving access to medical care for populations in the sub-region and scaling up possible management of diseases of concern to the sub-region.

Furthermore, although the current study reports on the landscape of herbal remedies employed for medicinal purposes in West Africa, the findings underscore the need for broader engagement and collaboration to expand the range of studies and countries engaged with trade-medicinal research, given its importance for complimentary healthcare delivery in the region, and the need to address neglected tropical diseases prevalent in the sub-region. Such collaborations will also enrich the diversity of perspectives and practices captured in research and facilitate the commercialization of validated products.

In conclusion, this scoping review mapping the modalities employed in traditional medicine research reveals a rich tapestry of temporal, geographical, and methodological diversity. The findings underscore the enduring relevance of traditional medicine across West Africa and emphasize the need for continued research to harness its full potential for healthcare. The study also highlights the need for further reviews of publications on traditional medicines written in other languages in the region to develop a rich compendium of the opportunities to harness the potential healthcare opportunities traditional medicines offer in the sub-region.

## Supporting information

**S1 Checklist. Preferred Reporting Items for Systematic reviews and Meta-Analyses extension for Scoping Reviews (PRISMA-ScR) checklist.**
(DOCX)

**S1 File. Search strategy for the CINAHL, web of science and pubmed.**
(DOCX)

**S2 File. A compendium of plant families used for medicinal purposes in West Africa.**
(DOCX)

**S3 File. Compendium of plant families used for experimentation.**
(DOCX)

## Author Contributions

**Conceptualization:** Selassi A. D'Almeida, Morẹ́nikẹ́ Oluwátóyìn Foláyan.

**Data curation:** Mobolaji Timothy Olagunju, Olunike Rebecca Abodunrin, Morẹ́nikẹ́ Oluwátóyìn Foláyan.

**Formal analysis:** Mobolaji Timothy Olagunju, Olunike Rebecca Abodunrin, Morẹ́nikẹ́ Oluwátóyìn Foláyan.

**Methodology:** Mobolaji Timothy Olagunju, Olunike Rebecca Abodunrin, Morẹ́nikẹ́ Oluwátóyìn Foláyan.

**Project administration:** Morẹ́nikẹ́ Oluwátóyìn Foláyan.

**Resources:** Sahr E. Gbomor, Brima Osaio-Kamara.

**Supervision:** Selassi A. D'Almeida, Morẹ́nikẹ́ Oluwátóyìn Foláyan.

**Validation:** Mobolaji Timothy Olagunju, Olunike Rebecca Abodunrin.

**Writing – original draft:** Morẹ́nikẹ́ Oluwátóyìn Foláyan.

**Writing – review & editing:** Selassi A. D'Almeida, Sahr E. Gbomor, Brima Osaio-Kamara, Mobolaji Timothy Olagunju, Olunike Rebecca Abodunrin, Morẹ́nikẹ́ Oluwátóyìn Foláyan.

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
