## [Decision Letter · Decision Letter 0]

26 Feb 2024

PONE-D-23-43939A scoping review on the use of traditional medicine for the management of ailments in West AfricaPLOS ONE

Dear Dr. Foláyan,

Thank you for submitting your manuscript to PLOS ONE. After careful consideration, we feel that it has merit but does not fully meet PLOS ONE’s publication criteria as it currently stands. Therefore, we invite you to submit a revised version of the manuscript that addresses the points raised during the review process.

We look forward to receiving your revised manuscript.

Kind regards,

Timothy Omara, PhD

Academic Editor

PLOS ONE

Journal Requirements:

2. Please amend your list of authors on the manuscript to ensure that each author is linked to an affiliation. Authors’ affiliations should reflect the institution where the work was done (if authors moved subsequently, you can also list the new affiliation stating “current affiliation:….” as necessary).

Additional Editor Comments:

In addition to the reviewer comments,

1. In your results section of the abstract, the occurrence of publications from 1979 to 2023 does not in essence attest to an increasing interest in traditional medicine unless if more studies were done in the last decade. This could have been fueled by the mounting resistance of pathogens, parasites and cancer cells among others to conventional medicine, and hence the expansive search for novel bioactive molecules.

2. As pointed out by one of the reviewers, it is less likely that Euphorbiaceae, Rubiaceae, and Combretaceae could have been the most documented/studied. To my knowledge, Fabaceae tends to be the most encountered in most studies, due to the abundant distribution of analogue active substances (flavonoids, terpenoids, and alkaloids) among species from this family.

3. The INTRODUCTION needs to be rethought about, so that it does not make the report too regional.

4. I have had troubles with the methodology used in this scoping review, and I expect that it could have introduced a potential error in the number of publications included.

(a) I am not quite sure that all studies included in PubMed, Web of Science, and CINAHL only could have given a good coverage of studies indexed in other multidisciplinary electronic databases like Scopus, Google Scholar, and Science Direct. There are new studies published in West Africa which does not appear in this report. For example,

Dossou, A. J., Fandohan, A. B., Djossa, A. B., & Assogbadjo, A. E. (2021). Diversity and knowledge of plants used in the treatment of snake bite envenomation in Benin. Ethnobotany Research and Applications, 21, 1-20.

Mengome, L. E., Mewono, L., Mboma, R., Engohang-Ndong, J., & Angone, S. A. (2021). Ethnobotanical survey and phytochemical screening of anti-snakebite plants used in Bissok District of Gabon. Biodiversitas Journal of Biological Diversity, 22(8).

Fanou, B. A., Klotoe, J. R., Fah, L., Dougnon, V., Koudokpon, C. H., Toko, G., & Loko, F. (2020). Ethnobotanical survey on plants used in the treatment of candidiasis in traditional markets of southern Benin. BMC Complementary Medicine and Therapies, 20, 1-18.

Please also check to include PubMed among the databases used in the main text METHODS.

(b) Why were only studies in English included? French is one of the widely spoken languages in West African countries such as Senegal, Mali, Guinea Conakry, Mauritania, Cote d'Ivoire, Burkina Faso, Togo, Benin, and Niger. There are indeed publications in French on this subject area.

5. The reporting needs to be improved. There are probably excessive in-text citations, and I suggest that you could take a glance at previous studies to help you visualize the retrieved data and improve your reporting.

Plaatjie et al. (2024) A scoping review on efficacy and safety of medicinal plants used for the treatment of diarrhea in sub-Saharan Africa. Trop Med Health 52, 6. https://doi.org/10.1186/s41182-023-00569-x

Mwaka et al. (2020). Traditional and Complementary Medicine Use Among Adult Cancer Patients Undergoing Conventional Treatment in Sub-Saharan Africa: A Scoping Review on the Use, Safety and Risks. Cancer management and research, 12, 3699–3712. https://doi.org/10.2147/CMAR.S251975

Obakiro et al. (2020). Ethnobotany, ethnopharmacology, and phytochemistry of traditional medicinal plants used in the management of symptoms of tuberculosis in East Africa: a systematic review. Tropical medicine and health, 48, 68. https://doi.org/10.1186/s41182-020-00256-1

6. Your DISCUSSION needs to be expanded. For example, how does the current data compare with other regions of Africa or other parts of the world?

Reviewers' comments:

Reviewer's Responses to Questions

**Comments to the Author**

1. Is the manuscript technically sound, and do the data support the conclusions?

Reviewer #1: Partly

Reviewer #2: Yes

2. Has the statistical analysis been performed appropriately and rigorously? 

Reviewer #1: Yes

Reviewer #2: Yes

3. Have the authors made all data underlying the findings in their manuscript fully available?

Reviewer #1: Yes

Reviewer #2: Yes

4. Is the manuscript presented in an intelligible fashion and written in standard English?

Reviewer #1: Yes

Reviewer #2: No

5. Review Comments to the Author

Reviewer #1: The manuscript is a scoping review of traditional medicine used in West Africa for the management of ailments.

The review is generally well-written. However, I have a major concern about the inclusion of ethnopharmacological/Experimental studies in the review, as the paper's objective is on traditional medicine.

Since the paper included ethnopharmacological/Experimental studies, I expected that the paper should shed more light on the extent to which traditional medicines have been validated in Africa; this should be part of the paper's objective to show the efficacy of traditional medicine.

There was also no critical assessment of the findings. For example, which plant part is mostly used in traditional medicine, and why do people tend to use it? This can be put in figures such as bar charts or pie charts. This could also be done for plant families, mode of administration and application of traditional medicine, and countries that depend mostly on traditional medicine.

I also doubt the assertion made by the authors that only Families Combretacea and Rubiaceae were reported in Benin and Mali, respectively, to be used in traditional medicine. For sure, other families are used in those countries. I also wonder why Family Fabaceae or Apocynaceae did not feature as one of the most studied or used plant families. These two plant families have emerged as the most used plants in many ethnobotanical studies around the globe.

I am also surprised that diabetics did not emerge as one of the ailments mostly treated with traditional medicine in West Africa. As far as I know, most ethnobotanical and ethnopharmacological studies in Nigeria have always focused on antidiabetic, antimicrobial and antioxidant studies.

Lastly, the authors should write all the scientific names in the manuscript in italics, as some scientific names in the text and tables are not properly written.

Reviewer #2: There are grammatical errors and incorrect presentation of scientific terms e.g. scientific names of a number of plant species and pathogens are not italicized and incorrectly written (wrong spelling and upper case on some species names). Details are on the attached file.

6. PLOS authors have the option to publish the peer review history of their article (what does this mean?). If published, this will include your full peer review and any attached files.

Reviewer #1: No

Reviewer #2: **Yes: **Denis Okello

---

## [Author Response · Author response to Decision Letter 0]

16 May 2024

A scoping review on the use of traditional medicine for the management of ailments in West Africa

PONE-D-23-43939

10th May 2024

We want to thank the reviewers for the painstaking efforts to review the manuscript and give constructive comments. Find below a point-by-point response to the reviewers’ comments

Editor’s comments

1. In your results section of the abstract, the occurrence of publications from 1979 to 2023 does not in essence attest to an increasing interest in traditional medicine unless if more studies were done in the last decade. This could have been fueled by the mounting resistance of pathogens, parasites and cancer cells among others to conventional medicine, and hence the expansive search for novel bioactive molecules.

RESPONSE: Thanks for this observation We have revised the statement, and it now reads: The search identified 3484 records, with 46 articles meeting inclusion criteria. Publications spanned from 1979 to 2023, with no observed trend in the number of publications with successive decades.

2. As pointed out by one of the reviewers, it is less likely that Euphorbiaceae, Rubiaceae, and Combretaceae could have been the most documented/studied. To my knowledge, Fabaceae tends to be the most encountered in most studies, due to the abundant distribution of analogue active substances (flavonoids, terpenoids, and alkaloids) among species from this family.

RESPONSE: Thank you for your observation. The discovery was derived from the document presented in Appendix 2 and Table 2. Our analysis was conducted on the evidence extracted from the reviewed manuscripts, leading to our conclusions. Additionally, we have acknowledged the study limitations, recognizing that our review was confined to manuscripts published in English and searches conducted in three databases, potentially restricting the comprehensive assessment of publications from West Africa. Nonetheless, we objectively evaluated the accessible publications and reported our findings accordingly. 

3. The INTRODUCTION needs to be rethought about, so that it does not make the report too regional.

RESPONSE: Thanks for the suggestion. We had focused on West Africa because this was the subject and context of the review. We have addressed this comment by expanding on the application of the study findings beyond West Africa in the discussion section. We wrote. Furthermore, although the current study reports on the landscape of herbal remedies employed for medicinal purposes in West Africa, the findings underscore the need for broader engagement and collaboration to expand the range of studies and countries engaged with trado-medicinal research in view of its importance for complimentary healthcare delivery in the region, and the need to address neglected tropical diseases prevalent in the sub-region. Such collaborations will also enrich the diversity of perspectives and practices captured in research and facilitate the commercialization of validated products.

4. I have had troubles with the methodology used in this scoping review, and I expect that it could have introduced a potential error in the number of publications included. 

(a) I am not quite sure that all studies included in PubMed, Web of Science, and CINAHL only could have given a good coverage of studies indexed in other multidisciplinary electronic databases like Scopus, Google Scholar, and Science Direct. There are new studies published in West Africa which does not appear in this report. For example,

Dossou, A. J., Fandohan, A. B., Djossa, A. B., & Assogbadjo, A. E. (2021). Diversity and knowledge of plants used in the treatment of snake bite envenomation in Benin. Ethnobotany Research and Applications, 21, 1-20.

Mengome, L. E., Mewono, L., Mboma, R., Engohang-Ndong, J., & Angone, S. A. (2021). Ethnobotanical survey and phytochemical screening of anti-snakebite plants used in Bissok District of Gabon. Biodiversitas Journal of Biological Diversity, 22(8).

Fanou, B. A., Klotoe, J. R., Fah, L., Dougnon, V., Koudokpon, C. H., Toko, G., & Loko, F. (2020). Ethnobotanical survey on plants used in the treatment of candidiasis in traditional markets of southern Benin. BMC Complementary Medicine and Therapies, 20, 1-18.

Please also check to include PubMed among the databases used in the main text METHODS.

RESPONSE: Thank you for the comprehensive information provided. The scoping review followed the prescribed procedure by Joanna Briggs Institute (JBI) scoping review methodology [11] and reported following the Preferred Reporting Items for Systematic Reviews and Meta-Analyses Extension for Scoping Reviews guidelines (PRISMA-ScR) [12,13]. As mentioned in the manuscript, data from Pubmed was indeed included in the database. The JBI guidelines suggest commencing with an initial search of at least two databases pertinent to the review topic, followed by an analysis of both the text words within the title and abstract, as well as the subject headings utilized in the retrieved papers. Generally, two to three databases are searched for a scoping review. We conducted our search in PubMed, Web of Science, and CINAHL thereby meeting the minimum standard for the scoping review. In view of the comments raised, we went back to review the studies extracted from the database. We have included the three studies are recommendation from the peer reviewer who is an expert in the field because if the search engine is run the three manuscripts would not be found in the list of publications enumerated. We are grateful for the insight and guidance. 

(b) Why were only studies in English included? French is one of the widely spoken languages in West African countries such as Senegal, Mali, Guinea Conakry, Mauritania, Cote d'Ivoire, Burkina Faso, Togo, Benin, and Niger. There are indeed publications in French on this subject area.

RESPONSE: We agree with you on the limitation of the use of English publications alone for a study in a region that has more French than English speaking countries. We have delayed the revision of this manuscript because of the need to find a collaborator who could do a scoping review with us who could also speak French. We have therefore acknowledged the limitation of the study by editing the title to reflect that the study was limited to publications from West Africa written in English. 

5. The reporting needs to be improved. There are probably excessive in-text citations, and I suggest that you could take a glance at previous studies to help you visualize the retrieved data and improve your reporting.

Plaatjie et al. (2024) A scoping review on efficacy and safety of medicinal plants used for the treatment of diarrhea in sub-Saharan Africa. Trop Med Health 52, 6. https://doi.org/10.1186/s41182-023-00569-x

Mwaka et al. (2020). Traditional and Complementary Medicine Use Among Adult Cancer Patients Undergoing Conventional Treatment in Sub-Saharan Africa: A Scoping Review on the Use, Safety and Risks. Cancer management and research, 12, 3699–3712. https://doi.org/10.2147/CMAR.S251975

Obakiro et al. (2020). Ethnobotany, ethnopharmacology, and phytochemistry of traditional medicinal plants used in the management of symptoms of tuberculosis in East Africa: a systematic review. Tropical medicine and health, 48, 68. https://doi.org/10.1186/s41182-020-00256-1

RESPONSE: we than the reviewer for the guidance. We have retained the format of reporting on this study finding as there are references of published scoping reviews that have used this method, and we as a team have adopted this reporting method in all our scoping reviews. 

6. Your DISCUSSION needs to be expanded. For example, how does the current data compare with other regions of Africa or other parts of the world?

RESPONSE: Thanks for the guidance. We wrote: Such disease-specific studies on herbal remedies are growing. These include reviews on herbal therapies for diarrhoea in sub-Saharan Africa [67], and global use of traditional medicines for cancer [68]. These disease-specific studies generate evidence for evaluating the efficacy and safety of these herbal remedies, which is crucial for integrating traditional medicine into mainstream healthcare systems help inform medical decision-making based on empirical data. Further studies are needed to compare the disease-specific herbal therapies used between regions, their efficacy and safety profile. 

Reviewer #1: The manuscript is a scoping review of traditional medicine used in West Africa for the management of ailments. The review is generally well-written. However, I have a major concern about the inclusion of ethnopharmacological/Experimental studies in the review, as the paper's objective is on traditional medicine.

RESPONSE: Thanks for the feedback. As indicated, this was a scoping review to highlight what had been done on traditional medicines in West Africa. The scoping review indicated that studies on traditional medicines had focused on ethnopharmacological/Experimental studies as well as ethnobotanical studies. What we did was highlight the findings from the mapping.

Since the paper included ethnopharmacological/Experimental studies, I expected that the paper should shed more light on the extent to which traditional medicines have been validated in Africa; this should be part of the paper's objective to show the efficacy of traditional medicine.

RESPONSE: Thanks for this suggestion. The study was a scoping review. All a scoping review can do it map the literature. We are not able to show the efficacy of the traditional medicines as we did not conduct a systematic review. Since this was a scoping review, a risk of bias for the study was not conducted. Attempting to a a study on efficacy of the traditional medicines will be outside the scope of a scoping review. For this reason, this was not conducted.

There was also no critical assessment of the findings. For example, which plant part is mostly used in traditional medicine, and why do people tend to use it? This can be put in figures such as bar charts or pie charts. This could also be done for plant families, mode of administration and application of traditional medicine, and countries that depend mostly on traditional medicine.

RESPONSE: Thanks for this insightful comment. We have now drawn a pie chart (Figure 2) for the parts of the plans used for traditional medicines. We wrote: The Figure 2 shows that leaves and roots are the most used parts for medicinal purposes, followed by barks and stems. Other parts such as fruits, seeds, and whole plants also contribute to a significant portion of plant-based remedies. We had done an analysis about the plant families as represented by the supplemental files 2 and 3. The mode of administration of the remedies have now been represented by an histograph (Figure 3). We have also discussed the findings further in the discussion section. We wrote: As indicated in the current study, the medicinal properties sourced from plants can originate from various plant components such as leaves, roots, bark, fruits, seeds, and flowers. Each part of a plant may harbor distinct active ingredients, contributing to its therapeutic effects. We found that leaves and roots are the most used parts for medicinal purposes in the region. Leaves, often rich in bioactive compounds, are frequently utilized for their therapeutic properties in various remedies and are prepared in a various forms ranging from decoctions to poultices. Roots are also valued for their medicinal properties and are often prepared as decoctions or infusions to extract their therapeutic properties. The range of plant parts used for medicinal purposes and the forms they are used in the current study reflects the diverse traditional knowledge and practices embedded in local communities across the West Africa.

I also doubt the assertion made by the authors that only Families Combretacea and Rubiaceae were reported in Benin and Mali, respectively, to be used in traditional medicine. For sure, other families are used in those countries. I also wonder why Family Fabaceae or Apocynaceae did not feature as one of the most studied or used plant families. These two plant families have emerged as the most used plants in many ethnobotanical studies around the globe.

RESPONSE: Thanks for the report. As indicated we mapped the findings in the study we did not make an assertion but rather, reported what we found in the study review. The families Combretacea and and Rubiaceae reported for Benin and Mali are those we found reported for experimental purposes. This does not preclude other families been used for other purposes as seen in table 2. A review of table 2 shows that other plant families are used in Benin and Mali. We have a compendium of plant families reported in Supplemental files 2 and 3. 

I am also surprised that diabetics did not emerge as one of the ailments mostly treated with traditional medicine in West Africa. As far as I know, most ethnobotanical and ethnopharmacological studies in Nigeria have always focused on antidiabetic, antimicrobial and antioxidant studies.

RESPONSE: Thanks for the observation. We have reported as objectively as possible the findings from the studies screened from the three databases used for this study. Diabetes was identified as diseases managed by traditional medicines in West Africa and we wrote: The remedies are also applicable to specific diseases like ----- diabetes [24, 27, 58] ----. 

Lastly, the authors should write all the scientific names in the manuscript in italics, as some scientific names in the text and tables are not properly written.

RESPONSE: Thanks for this observation. We have effected the suggested correction.

Reviewer #2

There are grammatical errors and incorrect presentation of scientific terms e.g. scientific names of a number of plant species and pathogens are not italicized and incorrectly written (wrong spelling and upper case on some species names). Details are on the attached file.

RESPONSE: Thanks for this guidance. We have corrected the italics. We have taken care to edit the entire manuscript for these errors and hope we have captured all the edits. We do not have access to the attached file.

The manuscript titled “A scoping review on the use of traditional medicine for the management of ailments in West Africa” presented a comprehensive overview of traditional healing practices in West Africa. It is an important study as it adds knowledge to the field of complementary and alternative medicine. The data search was generally well executed and methods clear. However, there are major issues with especially the presentation of results; I can only recommend publication of the work in this journal when authors make substantial amendments to the manuscript in the different sections as per these recommendations.

RESPONSE: Thanks for the feedback. We have painstakingly taken time to address the comments. We hope the responses prove satisfactory. 

Title page: match authors correctly to their affiliations; i do not see authors affiliated to 1, 2, 3.

RESPONSE: Apologies for the errors. This has been addressed. 

Abstract: arrange words alphabetically

RESPONSE: We arranged the study design and the plan families alphabetically. 

Introduction: correct the grammar in highlighted section 

RESPONSE: We have edited the introduction to improve the clarity. We do not have access to the document highlighted. We hope the revision done would have address the reviewer’s concern. 

Methods: correct grammar in highlighted sentences; replace appendix with supplementary file here and throughout the manuscript; specify data analysis software used 

RESPONSE: We have red through the manuscript to effect grammatical corrections. We have also used a software to correct the grammar. We did not have access to the attached document. We have replaced appendix with supplemental files and edited the legends accordingly. A data analysis soft ware was not used for the study. 

Results: be consistent in using figures or words as highlighted in different sections; table 1- All scientific names of all plants and some pathogens in the table are incorrectly written-not italicized and in some cases species names are started in uppercase. Go through the entire table carefully and correct these; You have mixed hyp

---

## [Decision Letter · Decision Letter 1]

2 Jun 2024

PONE-D-23-43939R1A scoping review of English publications on the use of traditional medicine for the management of ailments in West AfricaPLOS ONE

Dear Dr. Foláyan,

Thank you for submitting your manuscript to PLOS ONE. After careful consideration, we feel that it has merit but does not fully meet PLOS ONE’s publication criteria as it currently stands. Therefore, we invite you to submit a revised version of the manuscript that addresses the points raised during the review process.

We look forward to receiving your revised manuscript.

Kind regards,

Timothy Omara, PhD

Academic Editor

PLOS ONE

Journal Requirements:

Reviewers' comments:

Reviewer's Responses to Questions

**Comments to the Author**

1. If the authors have adequately addressed your comments raised in a previous round of review and you feel that this manuscript is now acceptable for publication, you may indicate that here to bypass the “Comments to the Author” section, enter your conflict of interest statement in the “Confidential to Editor” section, and submit your "Accept" recommendation.

Reviewer #1: All comments have been addressed

Reviewer #2: (No Response)

2. Is the manuscript technically sound, and do the data support the conclusions?

Reviewer #1: Yes

Reviewer #2: Yes

3. Has the statistical analysis been performed appropriately and rigorously? 

Reviewer #1: Yes

Reviewer #2: Yes

4. Have the authors made all data underlying the findings in their manuscript fully available?

Reviewer #1: Yes

Reviewer #2: Yes

5. Is the manuscript presented in an intelligible fashion and written in standard English?

Reviewer #1: No

Reviewer #2: No

6. Review Comments to the Author

Reviewer #1: The manuscript is a revised edition of a scoping review of traditional West African medicine for managing ailments. The manuscript has been significantly improved, and the authors have addressed my concerns. However, the title needs revision. There is no need for the authors to add 'English publications' to the title. The information should only be added to the inclusion criteria that the review is based on English publications.

Reviewer #2: Some efforts have been made by the authors to improve the quality of the manuscript based on the reviewers’ comments. However, there are still some grammatical errors in the document. I suggest the authors have the manuscript subjected to English editing and proofreading by institutions that offer these services.

Some of the issues that still need to be addressed by the authors in the different sections are here highlighted.

Abstract:

Key words must be arranged alphabetically

Introduction:

The sentence “This health-seeking behaviors across the region creating a nuanced healthcare ecosystem [6] and underscores the importance of acknowledging diverse healthcare practices and fostering collaborative efforts between traditional healers and modern healthcare professionals [7, 8].” is incorrect grammatically. Phrases used are pertinent to those generated by IA tools. Please rephrase and correct grammar.

Methodology:

In this sentence “The review was guided by the research questions: What is the extent and nature of Englishlanguage publications on the utilization of traditional medicine for treating various ailments in

West Africa.”

You talk of questions but I only see one.

In this sentence “A systematic search of the literature was conducted in PubMed, Web of Science, and the Cumulative Index of Nursing and Allied Health Literature (CINAHL) using the terms shown in S1 File.”

At first mention, write “S1 File” fully.

In this sentence “Thereafter, duplicate publications were removed. Three researchers (ORA, MTO, MOF) performed a screening of the titles and abstracts of the downloaded articles independently using pre-defined inclusion and exclusion criteria.”

write as “performed screening” and not” performed a screening”

In this sentence “Studies were included if there was an agreement between all the reviewers”

Use “among” instead of “between”. I don’t suppose the agreement was amongst reviewers but rather authors.

In this sentence “Animal studies were excluded. Also excluded were studies on non-African populations. Unpublished theses and dissertations, letters to the editor, commentaries on studies, scoping, systematic and narrative reviews, and studies whose full lengths cannot be accessed were excluded. Also, studies with insufficient results suitable for analysis were also excluded.”

Replace “cannot be accessed” with “could not be accessed”

In this sentence “As shown in Table 1, the publication period for the 49 manuscripts spanned from 1979 to 2023. The distribution across time revealed four (8.2%) were published between 1979 and 1990 [31- 33, 35], 20 (40.8%) were published between 1991 and 2000 [14-19, 20-23, 25, 38, 41-43, 46, 49, 51, 57, 59], 13 (26.5%) were from 2001 to 2010 [26, 36, 37, 40, 44, 47, 48, 50, 52-56], seven (14.3%) papers from 2011 to 2020 [27-30, 39, 58, 62], and five (10.2%) papers from 2021 to 2023 [24, 34, 45, 59, 61].”

you have mixed use of figures and words e.g. “revealed four (8.2%)” and “20 (40.8%) were published”. Please harmonize here and all through the results section.

Results

in table 1, there are a numbers of plant family names italicized. This is incorrect; please italicize only Genus and Species names. e.g. “Antidiarrhoeal Activities of Ocimum gratissimum (Lamiaceae)” do not italicize “Lamiaceae.” Do this throughout the document.

Throughout the texts in most cases family names have been italicized; please do not italicize. Do this throughout table 2.

Discussion

“Leaves, often rich in bioactive compounds, are frequently utilized for their therapeutic properties in various remedies and are prepared in a various forms ranging from decoctions to poultices. Roots are also valued for their medicinal properties and are often prepared as decoctions or infusions to extract their therapeutic properties. The range of plant parts used for medicinal purposes and the forms they are used in the current study reflects the diverse traditional knowledge and practices embedded in local communities across the West Africa”

Change “to extract their therapeutic properties” to “to extract their active ingredients”

7. PLOS authors have the option to publish the peer review history of their article (what does this mean?). If published, this will include your full peer review and any attached files.

Reviewer #1: No

Reviewer #2: **Yes: **Denis Okello

---

## [Author Response · Author response to Decision Letter 1]

7 Jun 2024

Title: A scoping review of the use of traditional medicine for the management of ailments in West Africa

Journal: Plos One

Manuscript number: PONE-D-23-43939R1

Date of Review: 3rd June 2024

We would like to thank the reviewers for the thorough review. Please find below the point-by-point response to the reviewers’ comments. We also conducted a grammar check. All edits are highlighted in red in the manuscript. 

Reviewer #1

The manuscript is a revised edition of a scoping review of traditional West African medicine for managing ailments. The manuscript has been significantly improved, and the authors have addressed my concerns. However, the title needs revision. There is no need for the authors to add 'English publications' to the title. The information should only be added to the inclusion criteria that the review is based on English publications.

Response: Thanks for raising this point. This has been edited. The title now reads: A scoping review of the use of traditional medicine for the management of ailments in West Africa. The information on English is now in the body of the manuscript.

Reviewer #2

 Some efforts have been made by the authors to improve the quality of the manuscript based on the reviewers’ comments. However, there are still some grammatical errors in the document. I suggest the authors have the manuscript subjected to English editing and proofreading by institutions that offer these services. Some of the issues that still need to be addressed by the authors in the different sections are here highlighted.

Response: Thanks for the suggestion. We have asked an editor to please review the manuscript for corrections. All edits are highlighted in red.

Abstract:

Key words must be arranged alphabetically

Response: Suggested edit effected.

Introduction:

The sentence “This health-seeking behaviors across the region creating a nuanced healthcare ecosystem [6] and underscores the importance of acknowledging diverse healthcare practices and fostering collaborative efforts between traditional healers and modern healthcare professionals [7, 8].” is incorrect grammatically. Phrases used are pertinent to those generated by IA tools. Please rephrase and correct grammar.

Response: Thanks for pointing this out. We now wrote: This health-seeking behaviors is a common practice across Africa creating a nuanced healthcare ecosystem in the region [6]. This underscores the importance of acknowledging diverse healthcare practices and fostering collaborative efforts between practitioners, including fostering collaborations between traditional healers and modern healthcare professionals [7, 8].

Methodology:

In this sentence “The review was guided by the research questions: What is the extent and nature of English language publications on the utilization of traditional medicine for treating various ailments in

West Africa.” You talk of questions but I only see one.

Response: We have now removed the ‘s’. Thanks for identifying this error. 

In this sentence “A systematic search of the literature was conducted in PubMed, Web of Science, and the Cumulative Index of Nursing and Allied Health Literature (CINAHL) using the terms shown in S1 File.”

At first mention, write “S1 File” fully.

Response: This is now written in full.

In this sentence “Thereafter, duplicate publications were removed. Three researchers (ORA, MTO, MOF) performed a screening of the titles and abstracts of the downloaded articles independently using pre-defined inclusion and exclusion criteria.” Write as “performed screening” and not” performed a screening”

Response: This is now written out in full. 

In this sentence “Studies were included if there was an agreement between all the reviewers”

Use “among” instead of “between”. I don’t suppose the agreement was amongst reviewers but rather authors.

Response: Thanks for the suggested edits. We wrote: Studies were included if there was an agreement among the three researchers who performed the screening.

In this sentence “Animal studies were excluded. Also excluded were studies on non-African populations. Unpublished theses and dissertations, letters to the editor, commentaries on studies, scoping, systematic and narrative reviews, and studies whose full lengths cannot be accessed were excluded. Also, studies with insufficient results suitable for analysis were also excluded.”

Replace “cannot be accessed” with “could not be accessed”

Response: Suggested edit was effected

In this sentence “As shown in Table 1, the publication period for the 49 manuscripts spanned from 1979 to 2023. The distribution across time revealed four (8.2%) were published between 1979 and 1990 [31- 33, 35], 20 (40.8%) were published between 1991 and 2000 [14-19, 20-23, 25, 38, 41-43, 46, 49, 51, 57, 59], 13 (26.5%) were from 2001 to 2010 [26, 36, 37, 40, 44, 47, 48, 50, 52-56], seven (14.3%) papers from 2011 to 2020 [27-30, 39, 58, 62], and five (10.2%) papers from 2021 to 2023 [24, 34, 45, 59, 61].”

you have mixed use of figures and words e.g. “revealed four (8.2%)” and “20 (40.8%) were published”. Please harmonize here and all through the results section.

Response: Thanks for raising this. We have left the mix of figures and words in line with requirements that figures below 10 should be written in words. When figures are in double digits, these can we written in figures.

Results

in table 1, there are a numbers of plant family names italicized. This is incorrect; please italicize only Genus and Species names. e.g. “Antidiarrhoeal Activities of Ocimum gratissimum (Lamiaceae)” do not italicize “Lamiaceae.” Do this throughout the document.

Response: Thanks for picking this up. Edits effected. 

Throughout the texts in most cases family names have been italicized; please do not italicize. Do this throughout table 2.

Response: Edits effected.

Discussion

“Leaves, often rich in bioactive compounds, are frequently utilized for their therapeutic properties in various remedies and are prepared in a various forms ranging from decoctions to poultices. Roots are also valued for their medicinal properties and are often prepared as decoctions or infusions to extract their therapeutic properties. The range of plant parts used for medicinal purposes and the forms they are used in the current study reflects the diverse traditional knowledge and practices embedded in local communities across the West Africa”. Change “to extract their therapeutic properties” to “to extract their active ingredients”

Response: Suggested edit effected.

---

## [Editor Report · Decision Letter 2]

21 Jun 2024

A scoping review of the use of traditional medicine for the management of ailments in West Africaa

PONE-D-23-43939R2

Dear Dr. Foláyan,

We’re pleased to inform you that your manuscript has been judged scientifically suitable for publication and will be formally accepted for publication once it meets all outstanding technical requirements.

Kind regards,

Timothy Omara, PhD

Academic Editor

PLOS ONE
---

## [Editor Report · Acceptance letter]

27 Jun 2024

PONE-D-23-43939R2 

PLOS ONE

Dear Dr. Foláyan, 

I'm pleased to inform you that your manuscript has been deemed suitable for publication in PLOS ONE. Congratulations! Your manuscript is now being handed over to our production team.

Kind regards, 

on behalf of

Dr. Timothy Omara 

Academic Editor

PLOS ONE